Subject Areas:
computational biology/biomathematics/
biophysics

Keywords:
morphogens, reaction–diffusion model,
morphogenesis, tissue size

Author for correspondence:
Osvaldo Chara
e-mails: osvaldo.chara@tu-dresden.de,
ochara@iflysib.unlp.edu.ar

# Size matters: tissue size as a marker for a transition between reaction–diffusion regimes in spatio-temporal distribution of morphogens

Alberto S. Ceccarelli[1], Augusto Borges[1,2,3] and Osvaldo Chara[1,4,5]

[1]Systems Biology Group (SysBio), Institute of Physics of Liquids and Biological Systems (IFLySIB), National Scientific and Technical Research Council (CONICET), University of La Plata, La Plata, Argentina
[2]Research Unit of Sensory Biology & Organogenesis, Helmholtz Zentrum München, Munich, Germany
[3]Graduate School of Quantitative Biosciences (QBM), Munich, Germany
[4]Center for Information Services and High Performance Computing (ZIH), Technische Universität Dresden, Dresden, Germany
[5]Instituto de Tecnología, Universidad Argentina de la Empresa (UADE), Buenos Aires, Argentina

OC, 0000-0002-0868-2507

The reaction–diffusion model constitutes one of the most influential mathematical models to study distribution of morphogens in tissues. Despite its widespread use, the effect of finite tissue size on model-predicted spatio-temporal morphogen distributions has not been completely elucidated. In this study, we analytically investigated the spatio-temporal distributions of morphogens predicted by a reaction–diffusion model in a finite one-dimensional domain, as a proxy for a biological tissue, and compared it with the solution of the infinite-domain model. We explored the reduced parameter, the tissue length in units of a characteristic reaction–diffusion length, and identified two reaction–diffusion regimes separated by a crossover tissue size estimated in approximately three characteristic reaction–diffusion lengths. While above this crossover the infinite-domain model constitutes a good approximation, it breaks below this crossover, whereas the finite-domain model faithfully describes the entire parameter space. We evaluated whether the infinite-domain model renders accurate estimations of diffusion coefficients when fitted to finite spatial profiles, a procedure typically followed in fluorescence recovery after photobleaching (FRAP) experiments. We found that the infinite-domain model overestimates diffusion coefficients when the domain is smaller than the crossover tissue size. Thus, the crossover tissue size may be instrumental in selecting the suitable reaction–diffusion model to study tissue morphogenesis.

# 1. Introduction

In their transition towards maturity, tissues are crucially regulated by molecules known as morphogens, whose precise spatio-temporal distribution triggers the downstream changes in protein expression responsible for the exact differentiation patterns. Nevertheless, tissues are not only an inert scaffold upon which morphogens spread, but are also fully responsible for the morphogen uptake or their transformation by means of specific biochemical reactions. The problem of how a morphogen propagates over a tissue while it is being eliminated was mathematically encoded in the exquisite reaction–diffusion model by the great Alan Turing, who coined the 'morphogen' neologism to illustrate its character of 'form generator' [1].

The reaction–diffusion model constitutes one of the most influential quantitative approaches within developmental biology. From Turing's aforementioned seminal article and the study from Gierer & Meinhardt [2], a progressive wealth of reaction–diffusion models were developed, paving the way to become an essential and pivotal concept to understand tissue morphogenesis [3–6]. The model was extensively used to investigate distributions of morphogens in a variety of tissues and organisms such as *Drosophila melanogaster* wing imaginal disc [7], chick limb [8] and the stripe pattern of *Danio rerio* [9], among other examples.

Previous studies have analytically investigated this model assuming an infinite domain [10,11]. Although the model relied on the idea that the reaction–diffusion characteristic length of the morphogen pattern was reasonably smaller than the domain, it is clear that biological tissues do not entail infinite lengths. Other reports investigated the model assuming a finite domain by using numerical [7,12] and analytical approaches [13–16]. To our knowledge, the role played by the size of the domain in the spatio-temporal patterning predicted by this model has not yet been elucidated.

In this study, we present the analytical solution of a reaction–diffusion model describing *de novo* formation of a morphogen and its spread within a finite domain, as a proxy for a tissue. We analytically investigated the behaviour of the model, in terms of a reduced parameter, representing the tissue length in units of the characteristic reaction–diffusion length. We fully characterized the finite-domain model in terms of morphological aspects of the spatial distributions and the time to reach the steady state to finally compare them with the corresponding predictions from the infinite-domain model. We found a crossover tissue size above which both models coincide. Importantly, below this crossover size, the finite-domain model becomes a better approximation. Finally, we recreated a fluorescence recovery after photobleaching (FRAP) *in silico* experiment and found that the infinite-domain model renders a less accurate estimation of the morphogen diffusion coefficient, for tissues smaller than the crossover length.

# 2. Results

## 2.1. The reaction–diffusion model in the infinite domain

Here we briefly summarize the well-known reaction–diffusion model assuming an infinite domain and its analytic solution [10,11]. Within this model, it is assumed that the dynamics of the morphogen are faster than the proliferation rate of the tissue cells and, as a consequence, advective effects can be neglected. Otherwise, an advective term could be included in the model [17]. Since during the developmental process tissues usually organize along with a particular axis [18,19], this model is studied in a one-dimensional setting [10,11]. It is assumed that the morphogen concentration $C_1(x,t)$ can diffuse with a diffusion coefficient $D$ and is linearly degraded with a rate $k$:

$$\frac{\partial C_1(x,t)}{\partial t} = D\frac{\partial^2 C_1(x,t)}{\partial x^2} - kC_1(x,t). \tag{2.1}$$

In the equation above, the first term $D(\partial^2 C_1(x,t)/\partial x^2)$ is the 'diffusive' term while the second term $-kC_1(x,t)$ represents the 'reaction' term, which in this case is linear and simulates a degradation or an uptake process. It is considered that there is no morphogen at the beginning, that is, the initial condition is

$$C_1(x,t=0) = 0. \tag{2.2}$$

The only source of morphogen is a constant flux $q$ located at the origin, represented by the first boundary condition:

$$\frac{dC_1}{dx}(x=0,t) = -\frac{q}{D}. \tag{2.3}$$

This reaction–diffusion model is also known as synthesis–diffusion–degradation [20], diffusion-decay [16] or diffusion equation with linear degradation model [21], highlighting the role of the source as well as the particular reaction term.

In this model, it can be assumed that the spatial domain is infinite and there is a Dirichlet boundary condition at the tip of the tissue representing a sink absorbing the morphogen:

$$\lim_{x \to \infty} C_1(x,t) = 0. \tag{2.4}$$

Alternatively, a Neumann boundary condition can be assumed at the tip of the tissue:

$$\lim_{x \to \infty} \frac{dC_1}{dx}(x,t) = 0. \tag{2.4'}$$

This model was extensively investigated by other authors, and the solution, regardless of the boundary condition assumed at the tip, is [10,11,13]

$$C_1(x,t) = \frac{q}{\sqrt{Dk}} e^{-x/\sqrt{D/k}} \left[ 1 - \frac{1}{2} erfc\left( \sqrt{kt} - \frac{x}{2\sqrt{Dt}} \right) - \frac{1}{2} e^{2x/\sqrt{D/k}} erfc\left( \sqrt{kt} + \frac{x}{2\sqrt{Dt}} \right) \right]. \tag{2.5}$$

where $erfc(x)$ is the complementary error function.

Space and time variables can be rewritten in terms of the following dimensionless variables $\varepsilon = x/\sqrt{D/k}$ and $\tau = kt$. Consequently, the morphogen flux at the tissue origin can be rewritten as $S = q/\sqrt{Dk}$ and the concentration as $C(\varepsilon,\tau) = C_1(\varepsilon,\tau)/S$. With this nondimensionalization, model equations (equations (2.1–2.4)) take the form:

$$\frac{\partial C}{\partial \tau} = \frac{\partial^2 C}{\partial \varepsilon^2} - C \tag{2.6}$$

and

$$C(\varepsilon,\tau = 0) = 0. \tag{2.7}$$

Where the morphogen source at the tissue origin, in nondimensional units, $\varepsilon = 0$, is

$$\frac{dC}{d\varepsilon}(\varepsilon = 0, \tau) = -1. \tag{2.8}$$

And the morphogen sink and no flux boundary conditions at $\varepsilon$ tending to infinite in the nondimensionalized units are

$$\lim_{\varepsilon \to \infty} C(\varepsilon,\tau) = 0 \tag{2.9}$$

and

$$\lim_{\varepsilon \to \infty} \frac{dC}{d\varepsilon}(\varepsilon,\tau) = 0. \tag{2.9'}$$

Which leads to this solution:

$$C(\varepsilon,\tau) = e^{-\varepsilon} \left[ 1 - \frac{1}{2} erfc\left( \sqrt{\tau} - \frac{\varepsilon}{2\sqrt{\tau}} \right) - \frac{1}{2} e^{2\varepsilon} erfc\left( \sqrt{\tau} + \frac{\varepsilon}{2\sqrt{\tau}} \right) \right]. \tag{2.10}$$

## 2.2. The reaction–diffusion model in finite domains: an analytical solution

The previous model variant entails an infinite domain (equations (2.4), (2.4'), (2.9) and (2.9')). Since biological tissue sizes require a finite domain, we decided to replace the sink boundary condition imposed by equation (2.4) with:

$$C_1(x = L,t) = 0. \tag{2.11}$$

and the no flux boundary condition given by equation (2.4') with:

$$\frac{dC_1}{dx}(x = L,t) = 0. \tag{2.11'}$$

where $L$ is the length of the tissue.

We defined the quantity, $R = L/\sqrt{D/k}$ which is the only model parameter. This quantity represents the tissue length $L$ in units of the characteristic reaction–diffusion length $\lambda$, defined as $\lambda = \sqrt{D/k}$ [21,22].

Thus, the sink boundary condition at $\varepsilon = R$ for this model in nondimensionalized units is

$$C(\varepsilon = R, \tau) = 0. \tag{2.12}$$

while the no flux boundary condition at $\varepsilon = R$ in nondimensionalized units is:

$$\frac{dC}{d\varepsilon}(\varepsilon = R, t) = 0. \tag{2.12'}$$

These equations replace equation (2.9) and (2.9′) in the §2.1, assuming the finitude of the tissue.

We found the analytical solution of the general model for finite tissues while assuming a sink boundary condition at $\varepsilon = R$ (equations (2.6–2.8) and (2.12)) in the nondimensionalized units to be as follows (see electronic supplementary material for the demonstration):

$$C(\varepsilon, \tau) = \left( \frac{e^{-\varepsilon}}{1 + e^{-2R}} - \frac{e^{\varepsilon}}{1 + e^{2R}} \right) - \sum_{j=0}^{\infty} \frac{2}{R} \frac{\cos((j+1/2)\pi\varepsilon/R)}{((j+1/2)\pi/R)^2 + 1} e^{-[((j+1/2)\pi/R)^2 + 1]\tau}. \tag{2.13}$$

While for the no flux boundary condition at $\varepsilon = R$ (equations (2.6–2.8) and (2.12′)), the solution in nondimensionalized units is as follows:

$$C(\varepsilon, \tau) = -\left( \frac{e^{\varepsilon}}{1 - e^{2R}} + \frac{e^{-\varepsilon}}{e^{-2R} - 1} \right) - \frac{e^{-\tau}}{R} + \sum_{j=1}^{\infty} -\frac{2}{R} \frac{\cos(j\pi\varepsilon/R)}{(j\pi/R)^2 + 1} e^{-[(j\pi/R)^2 + 1]\tau}. \tag{2.13'}$$

This solution was previously obtained by Umulis [13].

Moreover, we also found the solution for the model assuming a fixed non-null concentration in $\varepsilon = 0$ and a null concentration in $\varepsilon = R$ (see electronic supplementary material). Finally, we solved the finite-domain model for different boundary conditions in two simple examples in two dimensions (see electronic supplementary material).

To further corroborate the analytical solutions, we implemented the model numerically, by using a finite differences scheme (see electronic supplementary material). Our results indicate that the analytical solutions are in agreement with numerical simulations, both in one and two dimensions (electronic supplementary material, figures S1 and S2).

## 2.3. Transient morphogen distributions are qualitatively different between the infinite-domain model and the finite-domain model when they are of the order of the characteristic length $\lambda$ or smaller

We decided to compare the reaction–diffusion model assuming a finite tissue versus an infinite domain. With the selected nondimensionalization, the latter does not have any free parameters. By contrast, the finite model has only one free parameter, $R$, which represents the tissue size in units of the characteristic length of the morphogen profile $\lambda$. By using our analytical solution for the model of finite tissues (equation (2.13)), we explored the predicted morphogen spatial profiles at different tissue sizes (i.e. varying $R$) and compared them with those calculated from the previously known solution assuming an infinite domain (equation (2.10)), at three different time points (figure 1). We observed that the morphogen concentrations predicted by the model assuming an infinite domain are higher or smaller than those predicted by the model assuming a finite domain, depending on the boundary condition assumed at $\varepsilon = R$ (figure 1a,b). For large enough tissue lengths, morphogen profiles predicted by both models are indistinguishable at each time point, as expected (figure 1c–f). Hence, the previously reported model assuming an infinite domain is a reasonable description of the dynamics of morphogen profiles for larger tissues. However, when addressing a tissue whose length is of the order of the characteristic length $\lambda$ or smaller, the model introduced in the present work is a more accurate description.

Moreover, we observed that large tissues lead to morphogen spatial distributions temporarily separated when assuming a sink boundary condition at $\varepsilon = R$. By contrast, spatial distributions at different time points are indistinguishable in shorter tissues, suggesting that they already approached the steady state (figure 1a). This result would indicate that the larger the tissue, the longer the time necessary to reach the morphogen spatial distribution at the steady state (see also sections 2.4 and 2.6). By contrast, assuming a no flux boundary condition at $\varepsilon = R$ leads to morphogen spatial

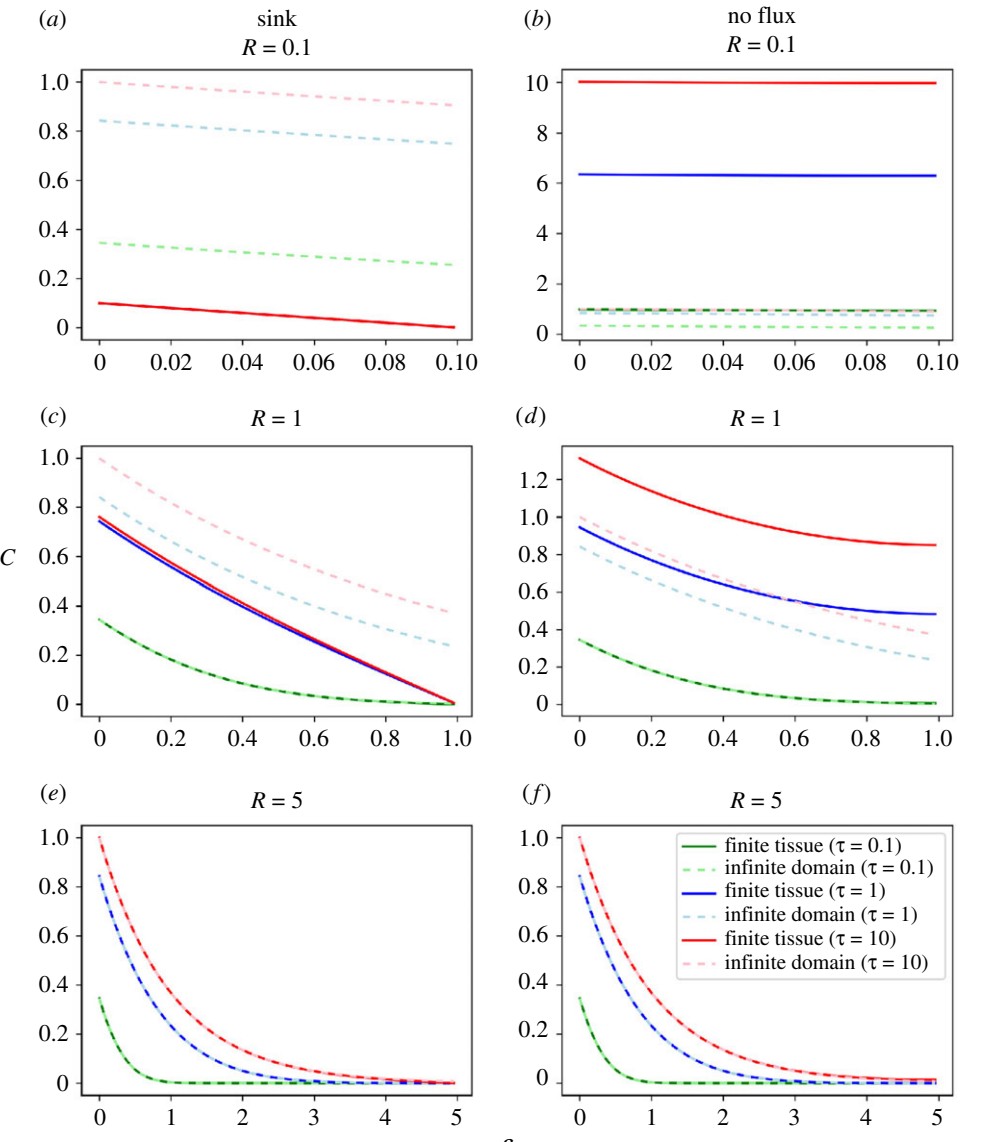

**Figure 1.** Morphogen spatial profiles predicted by the reaction–diffusion finite-domain model converge to the profile predicted by the infinite-domain model, for large tissue lengths. Morphogen spatial profiles of the reaction–diffusion finite-domain model at three nondimensionalized tissue sizes $R = 0.1$ (a,b), 1 (c,d) and 5 (e,f) and at three different nondimensionalized times $\tau = 0.1$, 1 and 10 (solid lines), assuming either a sink boundary condition (a,c and e) or a no flux boundary condition (b,d and f) at the nondimensionalized position $\varepsilon = R$. The profiles predicted by the model assuming an infinite domain are also shown at the same times (dashed lines). $C$ and $\varepsilon$ represent the nondimensionalized morphogen concentration and position, respectively.

distributions more temporally separated, suggesting that the time necessary to reach the steady state is longer in small tissues (see sections 2.4 and 2.6).

## 2.4. Steady state morphogen spatial distributions

The morphogen spatial distribution assuming an infinite domain at the steady state ($C_{ss}^{\text{infinite}}(\varepsilon)$) is well known [10,11] and with our nondimensionalization it is the following exponential spatial decay:

$$C_{ss}^{\text{infinite}}(\varepsilon) = e^{-\varepsilon}. \tag{2.14}$$

We calculated the steady state solution for our model of finite tissues, $C_{ss}^{\text{finite}}(\varepsilon)$, assuming a sink boundary condition at $\varepsilon = R$ (electronic supplementary material):

$$C_{ss}^{\text{finite}}(\varepsilon) = \left( \frac{e^{-\varepsilon}}{1 + e^{-2R}} - \frac{e^{\varepsilon}}{1 + e^{2R}} \right). \tag{2.15}$$

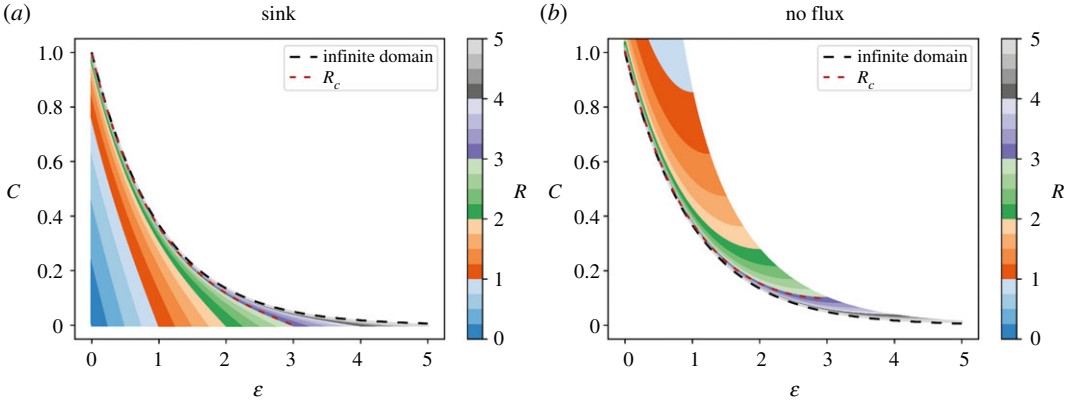

**Figure 2.** Steady state spatial profiles predicted by the reaction–diffusion finite-domain model converge to the profile predicted by the infinite-domain model, for large tissue lengths. Morphogen spatial profiles of the reaction–diffusion model at steady state assuming finite domains representing different nondimensionalized tissue lengths ($R$, colourbar) with a sink boundary condition (a) or a no flux boundary condition (b) at the nondimensionalized position $\varepsilon = R$. The steady state profile from the model assuming an infinite domain and the crossover tissue size $R_c$ (defined in §2.5) are shown as dashed black and red lines, respectively.

Increasing the tissue size in this model modifies the steady state profile, augmenting the maximum concentration at the origin and leading to a transition from a linear to an exponential curve (figure 2a), in agreement with the results observed at any time (figure 1a,c,e). Precisely, to estimate the limit when the tissue size tends to zero, we calculated the Taylor series expansion of the steady state solution (equation (2.15)) on $R$ to the first order. As $\varepsilon$ is constrained by $R$, we subsequently obtained the Taylor series expansion for the resulting expression on $\varepsilon$ to the first order:

$$\lim_{R \to 0} C_{SS}^{\text{finite}}(\varepsilon) \cong \lim_{\varepsilon \to 0} [-\sinh(\varepsilon) + R \cosh(\varepsilon)] \cong R - \varepsilon. \tag{2.16}$$

The limit when the tissue size tends to infinite was calculated:

$$\lim_{R \to \infty} C_{SS}^{\text{finite}}(\varepsilon) = e^{-\varepsilon} = C_{ss}^{\text{infinite}}(\varepsilon). \tag{2.17}$$

The steady state morphogen distribution of the finite model converges to the exponential distribution predicted by the infinite domain when the tissue length tends to infinity.

The model of finite tissues assuming no flux boundary condition at $\varepsilon = R$ has the following steady state solution:

$$C_{ss}^{\text{finite}}(\varepsilon) = -\left( \frac{e^{\varepsilon}}{1 - e^{2R}} + \frac{e^{-\varepsilon}}{e^{-2R} - 1} \right). \tag{2.15'}$$

With this boundary condition, the total amount of morphogen accumulated in the tissue at the steady state ($N_{SS}$) is conserved and consequently, independent of $R$ (see electronic supplementary material):

$$N_{ss} = \int_0^R C_{ss}^{\text{finite}}(\varepsilon) \, d\varepsilon = 1. \tag{2.18}$$

This result can be interpreted as follows. At the steady state, the morphogen net flux in the tissue must be equal to zero. There is only one morphogen influx given by the source, located at the origin, which is constant and equal to $-1$. Since the linear degradation term $-C$ (see equation (2.6)) is the only one responsible for morphogen depletion, the integral of this term over the entire tissue results in the morphogen efflux equal to $-N_{ss}$. Hence, the zero net flux requires that the influx balances out the efflux implying that $N_{ss} = 1$.

For small tissues, we calculated the first order of the Laurent series expansion of $C_{ss}^{\text{finite}}(\varepsilon)$ in $R = 0$ and, as $\varepsilon$ is constrained by $R$, we got $C_{ss}^{\text{finite}} \sim 1/R$ for small values of $\varepsilon$. Hence, the smaller the tissue, the higher the concentration averaged over the tissue. This explains why the steady concentration profiles of the finite tissues model assuming no flux at $\varepsilon = R$ are higher than the steady state profile of the infinite-domain model, in contrast with the result obtained with the sink boundary condition (figure 1b,d,f and figure 2). As expected, the steady state profile of the finite tissue model converges to

the spatial distribution predicted with the infinite-domain model when the tissue size tends to infinite (i.e. equation (2.17) holds for this boundary condition as well).

Furthermore, by comparing the steady state solution assuming a sink boundary condition at $\varepsilon = R$ (equation (2.15)) with its complete solution (equation (2.13)), we can re-write equation (2.13) as follows:

$$C(\varepsilon,\tau) = C_{ss}^{\text{finite}}(\varepsilon) - \sum_{j=0}^{\infty} \frac{2}{R} \frac{\cos((j+1/2)\pi\varepsilon/R)}{((j+1/2)\pi/R)^2 + 1} e^{-[((j+1/2)\pi/R)^2 + 1]\tau}. \tag{2.19}$$

Analogously, assuming no flux boundary condition at $\varepsilon = R$ leads to

$$C(\varepsilon,\tau) = C_{ss}^{\text{finite}}(\varepsilon) - \left[ \frac{e^{-\tau}}{R} + \sum_{j=1}^{\infty} \frac{2}{R} \frac{\cos(j\pi\varepsilon/R)}{(j\pi/R)^2 + 1} e^{-[(j\pi/R)^2 + 1]\tau} \right]. \tag{2.19$'$}$$

The second term of equations (2.19) and (2.19$'$) vanishes when the time $\tau$ tends to infinity. Therefore, the morphogen concentration can be expressed as the steady state solution plus a term that describes a transient contribution.

## 2.5. Geometrical characterization of the morphogen spatial distributions

The steady state profiles predicted by the model of finite tissues changed when increasing the tissue size as shown in §2.4. In order to geometrically characterize the shape of the spatial profiles in the steady state regime, we defined $\varepsilon_{10}$ as the dimensionless spatial position $\varepsilon$ in which the morphogen concentration is 10% of the concentration at the origin. When using this definition in the model assuming infinite domains, we obtain (see electronic supplementary material for details):

$$\varepsilon_{10} = ln(10) \cong 2.3. \tag{2.20}$$

While for the model of finite tissues assuming a sink boundary condition at $\varepsilon = R$ (see electronic supplementary material for details):

$$\varepsilon_{10} = R - \text{arc} \sinh\left(\frac{\sinh(R)}{10}\right). \tag{2.21}$$

Thus, in the limit of small tissues, $\varepsilon_{10}$ shows a linear dependence with the tissue size. However, when the tissue tends to infinity, $\varepsilon_{10}$ becomes independent of the precise tissue size, reaching a plateau (figure 3a). Additionally, when tissue size tends to infinity, in equation (2.21), $\varepsilon_{10}$ recovers the value from the infinite model calculated in equation (2.20).

We wonder whether it is possible to establish a cut-off size to distinguish both regimes. To answer this question, we explore under what conditions the shape of the morphogen spatial distribution depends on the tissue size. More precisely, we asked under what crossover tissue size $R_c$ the geometrical observable $\varepsilon_{10}$ would transition from linearly depending on the tissue size to becoming independent of it. To this end, we Taylor-expanded $\varepsilon_{10}$ and arbitrarily looked for the $R = R_c$ upon which the second non-zero term of the series would be about 20% of the first linear term (See electronic supplementary material for details). Our results show that the crossover tissue size separating both regimes is about three times the characteristic length $\lambda$ ($R_c \approx 3$). A similar analysis with the model of finite tissues assuming a no flux boundary condition at $\varepsilon = R$ renders a similar crossover tissue of about three $\lambda$ (figure 3b; electronic supplementary material).

The analysis of the dependency of $\varepsilon_{10}$ with the tissue size can also be made before the morphogen distribution achieves the steady state. Although we could not find an analytical expression for this observable in the general case, we explored this dependency numerically (figure 4). For both boundary conditions, we observed that $\varepsilon_{10}$ changes in time until it reaches a plateau, which indicates that the spatial profile stabilizes in the steady state. Moreover, for the sink boundary condition, the time needed to reach the plateau monotonically increases with the tissue size until $R \sim R_c$. By contrast, for the no flux boundary condition, the opposite is true: the time needed to reach the plateau monotonically decreases with the tissue size until $R \sim R_c$. For larger tissue sizes, the time to reach the plateau converges to the prediction of the model for infinite domains (figure 4a,b).

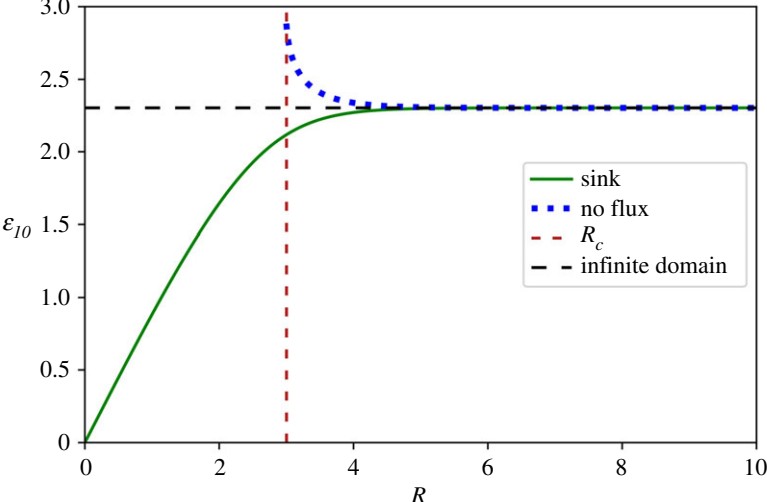

**Figure 3.** Crossover tissue size separating two reaction–diffusion model regimes. $\varepsilon_{10}$, defined as the nondimensionalized spatial position where the morphogen concentration is 10% of its value at the origin was calculated for the finite-domain model for different nondimensionalized tissue sizes ($R$), assuming a sink boundary condition (upper dotted blue curve) or a no flux boundary condition (lower continuous green curve) at the position $\varepsilon = R$. The finite-size model leads to $\varepsilon_{10}$ values different from those predicted by the infinite-domain model (horizontal discontinuous black curve) for tissue lengths smaller than the crossover tissue length ($R_c$, vertical discontinuous red curve): the sink (no flux) boundary condition results in $\varepsilon_{10}$ values smaller (higher) than the value predicted by the infinite-domain model. By contrast, $\varepsilon_{10}$ predicted by the finite-domain model is indistinguishable from the prediction of the infinite-domain model, for tissues larger than the crossover tissue length.

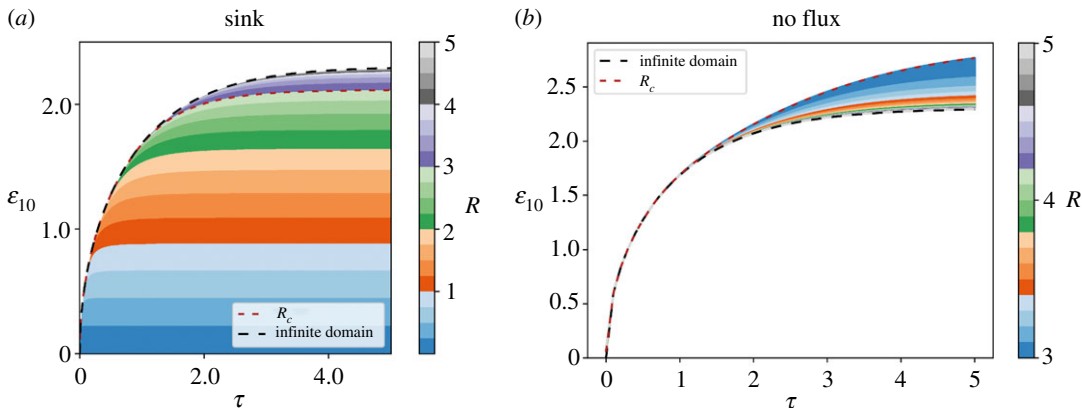

**Figure 4.** Kinetics of the geometrical factor $\varepsilon_{10}$ calculated from the reaction–diffusion finite-domain model for different tissue sizes. Kinetics of the geometrical factor $\varepsilon_{10}$ predicted from the finite-domain model for different nondimensionalized tissue sizes ($R$, tissue length in nondimensionalized units), assuming a sink boundary condition ($a$) or a no flux boundary condition ($b$) at the nondimensionalized position $\varepsilon = R$. The time course of the $\varepsilon_{10}$ factor corresponding to the infinite-domain model is also shown (dashed black line). $\varepsilon_{10}$ is defined as the spatial position where the morphogen concentration is 10% of its value at the origin. The dashed red line shows the crossover tissue length $R_c$.

## 2.6. Time to reach the steady state morphogen distribution

The results of §2.5 suggest that the larger the tissue, the longer (shorter) it takes the model to reach the steady state when assuming a sink (no flux) boundary condition at $e = R$. To test this hypothesis, we took advantage of a method developed by Berezhkovskii *et al.* [11] to quantify the mean time ($\mu_\tau$) it takes a morphogen profile to reach its steady state. In their study, the authors defined a local relaxation function as the ratio of the difference between the current and steady state values of the total amount of morphogen accumulated in the tissue. They interpret the negative derivative of the relaxation function as the probability density for the time of the local accumulation process from which they

could calculate the mean time. They applied this method to the reaction–diffusion model assuming an infinite domain and obtained (in our nondimensional units):

$$\mu_\tau(\varepsilon) = \frac{\varepsilon + 1}{2}. \tag{2.22}$$

That is, the mean time to reach the steady state is linear with the position within the infinite domain. We applied the same method to our reaction–diffusion model of finite tissues assuming a sink boundary condition at $\varepsilon = R$ and obtained (see electronic supplementary material for details):

$$\mu_\tau(\varepsilon) = \sum_{j=0}^{\infty} \frac{2}{R} \frac{\cos((j+1/2)\pi\varepsilon/R)}{[(((j+1/2)\pi/R))^2 + 1]^2} \frac{1}{(e^{-\varepsilon}/(1 + e^{-2R}) - e^{\varepsilon}/(1 + e^{2R}))}. \tag{2.23}$$

While assuming a no flux boundary condition at $\varepsilon = R$ renders (see electronic supplementary material for details):

$$\mu_\tau(\varepsilon) = \frac{1}{-(e^{\varepsilon}/(1 - e^{2R}) + e^{-\varepsilon}/(e^{-2R} - 1))} \left[ \frac{1}{R} + \sum_{j=1}^{\infty} \frac{2}{R} \frac{\cos(j\pi\varepsilon/R)}{((j\pi/R)^2 + 1)^2} \right]. \tag{2.23'}$$

Thus, for our model, the mean time to reach the steady state not only depends on the position within the tissue but also on the tissue size.

To formally compare the mean times calculated from both reaction–diffusion models, we also need to estimate a measure of the error. Hence, we calculated the standard deviation of the time to reach the steady state, $\sigma_\tau$ (see electronic supplementary material for details). For the model assuming an infinite domain, it reads:

$$\sigma_\tau(\varepsilon) = \frac{\sqrt{\varepsilon + 2}}{2}. \tag{2.24}$$

Which coincides with the already reported result by Ellery *et al.* [15]. By contrast, for the reaction–diffusion model for finite tissues assuming a sink boundary condition at $\varepsilon = R$, we obtain (see electronic supplementary material for details):

$$\sigma_\tau(\varepsilon) = \left( \sum_{j=0}^{\infty} \frac{4}{R} \frac{\cos((j+1/2)\pi\varepsilon/R)}{[(((j+1/2)\pi/R))^2 + 1]^3} \frac{1}{(e^{-\varepsilon}/(1 + e^{-2R}) - e^{\varepsilon}/(1 + e^{2R}))} \right.$$
$$\left. - \left( \sum_{j=0}^{\infty} \frac{2}{R} \frac{\cos((j+1/2)\pi\varepsilon/R)}{[(((j+1/2)\pi/R))^2 + 1]^2} \frac{1}{(e^{-\varepsilon}/(1 + e^{-2R}) - e^{\varepsilon}/(1 + e^{2R}))} \right)^2 \right)^{1/2}. \tag{2.25}$$

While assuming a no flux boundary condition at $\varepsilon = R$ renders (see electronic supplementary material for details):

$$\sigma_\tau(\varepsilon) = \left( \frac{2}{-(e^{\varepsilon}/1 - e^{2R}) + e^{-\varepsilon}/(e^{-2R} - 1))} \left[ \frac{1}{R} + \sum_{j=1}^{\infty} \frac{2}{R} \frac{\cos(j\pi\varepsilon/R)}{((j\pi/R)^2 + 1)^3} \right] \right.$$
$$\left. - \left( \frac{1}{-(e^{\varepsilon}/(1 - e^{2R}) + e^{-\varepsilon}/(e^{-2R} - 1))} \left[ \frac{1}{R} + \sum_{j=1}^{\infty} \frac{2}{R} \frac{\cos(j\pi\varepsilon/R)}{((j\pi/R)^2 + 1)^2} \right] \right)^2 \right)^{1/2}. \tag{2.25'}$$

As with the mean, the standard deviation of the time necessary to reach the steady state not only depends on the positions along with the tissue but also on the tissue size. At the origin of the tissue ($\varepsilon = 0$), both $\mu_\tau$ and $\sigma_\tau$ increase or decrease with $R$ (depending on whether there is a sink or a no flux boundary condition at $\varepsilon = R$) until they converge toward $1/2$ and $\sqrt{2}/2$, respectively, when $R$ tends to infinite (figure 5a). These are precisely the expected values from the model assuming infinite domains evaluated at the tissue origin (equations (2.22) and (2.24)). Interestingly, the transition between the domains in which $\mu_\tau$ and $\sigma_\tau$ depend on the tissue size and where they are independent of it coincides with the crossover tissue size of about three $\lambda$ determined in the previous section (compare figure 5a with figure 3).

For tissues smaller than the crossover size, the mean time to achieve the steady state and its error in each position strongly depend on tissue size (figure 5b). On the contrary, for tissue sizes higher than the crossover tissue size, both magnitudes become independent of the size (figure 5c). Importantly, for

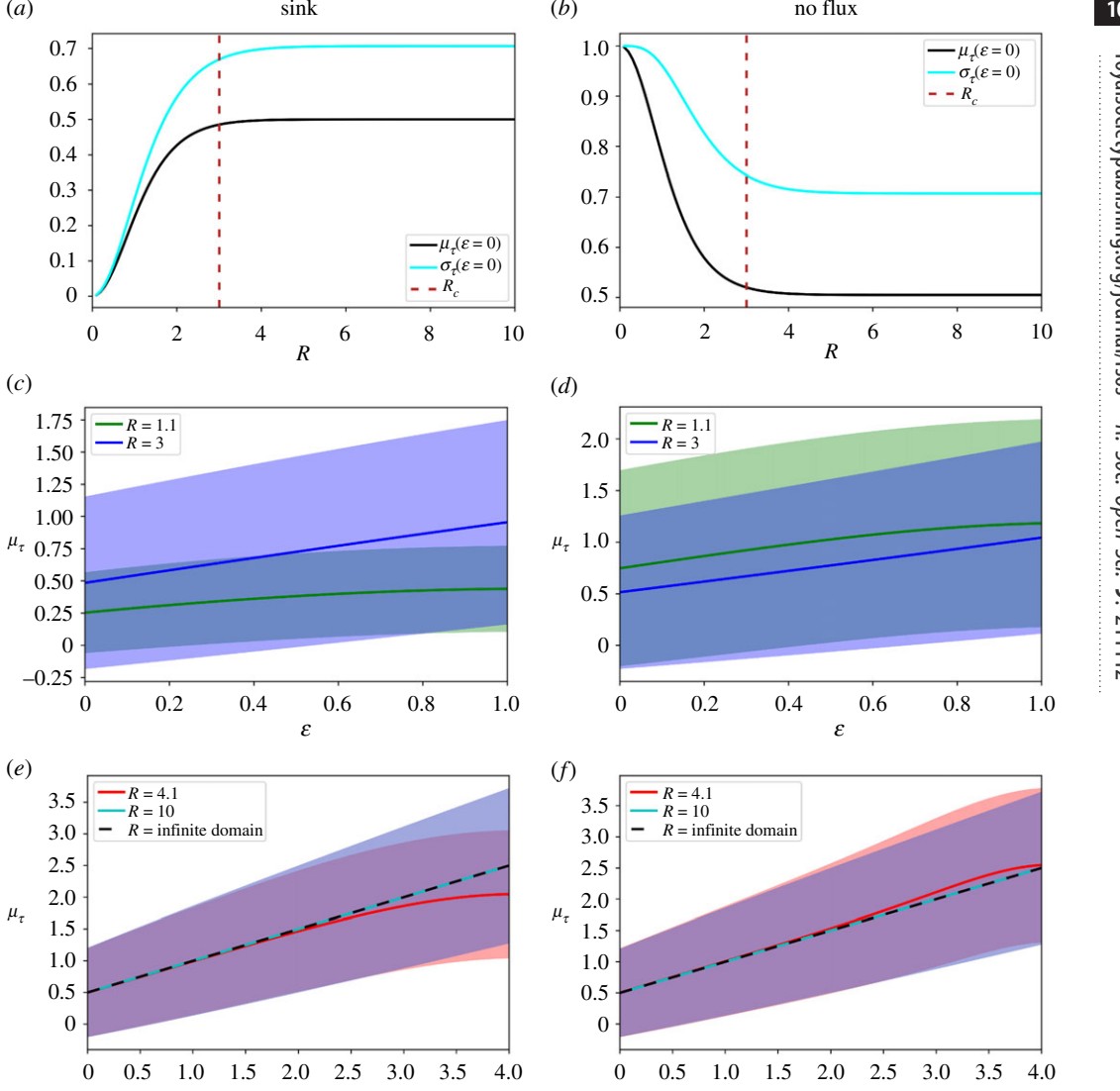

**Figure 5.** Crossover tissue size separates two regimes of kinetics to reach the steady state. (*a,b*) The mean time to reach the steady state ($\mu_\tau$, black line) and the standard deviation ($\sigma_\tau$, light blue line) at the nondimensionalized position $\varepsilon = 0$ as a function of the nondimensionalized tissue length ($R$) predicted by the reaction–diffusion finite-domain model assuming a sink (*a*) or a no flux boundary condition (*b*) at $\varepsilon = R$. The vertical dashed red line indicates the crossover tissue size $R_c$. (*c–f*) Spatial profiles of the mean time to reach the steady state for the finite-domain model assuming $R = 1.1$ (green line, *c,d*), 3 (blue line, *c,d*), 4.1 (red line, *e* and *f*) and 10 (light blue line, *e,f*). We assume a sink (*b,e*) or a no flux boundary condition (*d,f*) at $\varepsilon = R$. The standard deviation is represented by the shady areas surrounding the curves. The result corresponding to the infinite-domain model is shown in a dashed black line (*e,f*).

tissues smaller than the crossover size, the steady state will be reached significantly faster or slower than the prediction from the model assuming an infinite domain, depending on whether there is a sink or a no flux boundary condition, respectively, at $\varepsilon = R$. For tissues larger than the crossover size, both models agree in the time to achieve steady state (figure 5*b,c*).

## 2.7. Finite versus infinite domains in the reaction–diffusion model used in the fluorescence recovery after photobleaching-based determination of diffusion parameters

Diffusion parameters of morphogens can be experimentally determined in tissues by using fluorescence recovery after photobleaching (FRAP) experiments [23,24]. From this technique, the diffusion coefficient $D$ and degradation constant $k$ are obtained indirectly by fitting to experimental concentration measurements the analytical solution of the model assuming a finite domain [23,24] as well as an

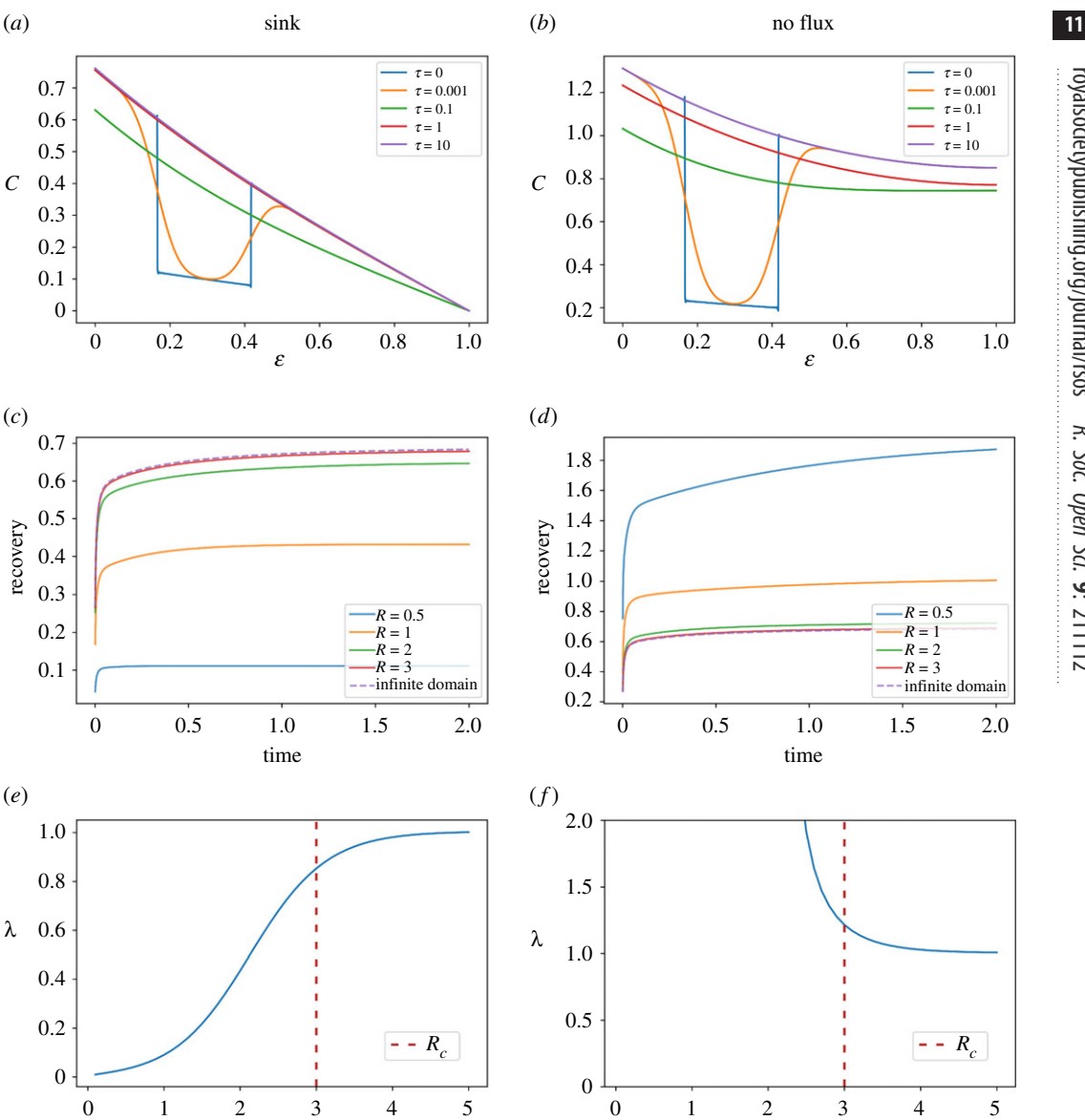

**Figure 6.** The reaction–diffusion model assuming an infinite domain does not correctly predict the characteristic length for tissues smaller than the crossover size in a simulated FRAP scenario. (a,b) Morphogen spatial profiles of the reaction–diffusion model in a finite domain of nondimensionalized tissue size $R = 1$ at different nondimensionalized times ($\tau$). At the initial condition ($\tau = 0$), the morphogen concentration is in the steady state except between the nondimensionalized positions $\varepsilon = 1/6$ and $\varepsilon = 5/12$, where the morphogen concentration is 0.2 times the concentration at steady state, to emulate a bleaching. (c,d) Simulated time evolution of the FRAP recovery curves predicted by the finite model for different tissue sizes (continuous lines) and compared with the infinite-domain model (discontinuous line). (e,f) Characteristic length ($\lambda$) predicted by the infinite-domain model when fitting the recovery curves generated by the finite-domain model where $\lambda$ was fixed in 1. Crossover tissue size ($R_c$) is indicated as the vertical discontinuous line in (e,f). The finite-domain model was simulated by assuming a sink boundary condition (a,c,e) or a no flux boundary condition (b,d,f) at $\varepsilon = R$.

infinite domain [19,25]. Thus, we wondered whether the election of the model has an impact on a typical FRAP scenario. To answer this question, we simulated a FRAP experiment with our finite-domain model by using as an initial condition the steady state distribution and recreated the bleaching by multiplying the steady state profile by an arbitrary factor $b$ between positions $d$ and $d + h$, following Kicheva *et al.* [19]. Of note, to model a FRAP experiment, a uniform distribution can be assumed before bleaching of the diffusing substance [25]. As our model entails a morphogen gradient generated by a source while undergoing diffusion and degradation, we focused our analysis on a non-uniform case. We next evaluated the morphogen spatio-temporal distribution predicted by the finite-domain model, assuming a sink and a no flux boundary condition at $\varepsilon = R$. As time passes, the morphogen gradient

recovered the steady state profile (figure 6$a$,$b$). We calculated a typical recovery curve defined as the spatial integral of the concentration between positions $d$ and $d + h$, divided by the length of the bleached zone $h$ (figure 6$c$,$d$). We calculated the recovery curves with our finite-domain model assuming either a sink or a no flux boundary condition at $\varepsilon = R$ (see equations S.31 and S.32, respectively, in the electronic supplementary material), and we compared them with the recovery curves predicted by the one-dimensional infinite-domain model, previously calculated by Kicheva et al. [19] (equation S.33 in the electronic supplementary material). Our results show that the recovery curves predicted by tissues larger than $R_c$ are indistinguishable from the curve predicted by the infinite-domain model (figure 6$c$,$d$). By contrast, tissues smaller than $R_c$ give rise to kinetics of recovery that differ from those predicted by the infinite-domain model. While a sink boundary condition at $\varepsilon = R$ gives rise to kinetics faster than those predicted by the infinite-domain model, the opposite occurs with the no flux boundary condition, in agreement with the results shown in the previous section.

Next, we wondered whether the election of the model used in FRAP has an impact on the calculated $D$ and $k$ values. To that end, as a proof of principle, we evaluated whether the infinite-domain model could render an accurate estimation of the kinetic parameters $D$ and $k$, when fitted to a recovery curve, in turn generated with the finite domain model used as a proxy for an experimental recovery curve.

After rewriting the recovery curve equations (see equations S.31 and S.32, respectively, in the electronic supplementary material) in the original coordinate $x = \lambda\varepsilon$ and arbitrarily setting $\lambda = 1$, we run simulations of the finite-domain model, for different values of $L$. Then, we performed a curve fitting for each of the simulated recovery curves by using the expression corresponding to the infinite-domain model as the fitting function and $\lambda$ as the free parameter.

We obtained the predicted value of $\lambda$ as a function of $R = L$ (figure 6$e$,$f$). For large values of $R$, the predicted $\lambda$ is approximately 1, which is in agreement with the value actually used to generate the data. By contrast, for values of $R$ smaller than $R_c$, the predicted value of $\lambda$ deviates from 1, converging to 0 or getting values orders of magnitude higher than 1, depending on the boundary condition at $\varepsilon = R$. Altogether, these results indicate that both models can be used to infer the kinetic parameters $D$ and $k$ from FRAP experiments, provided that tissue sizes are higher than $R_c$. On the contrary, for tissues smaller than this crossover value, the model assuming finite domains is the best alternative.

## 3. Discussion

Reaction–diffusion models were conceived in the seminal article by Alan Turing to hypothesize under what conditions heterogeneous patterns could emerge from a homogeneous one in tissue morphogenesis [1]. After the concept of positional information was posed by Lewis Wolpert [26], as illustrated by his well-known French Flag Problem ([27]; see also the review by Sharpe [28]), reaction–diffusion models resurfaced to account for mechanisms capable of generating spatial gradients that could serve as positional signals. Francis Crick was entertaining the hypothesis of reaction–diffusion signals as probable morphogenetic driving forces [29]. Reaction–diffusion models were specifically studied by Alfred Gierer and Hans Meinhardt to understand pattern formation in tissue development and regeneration [2]. Thereafter, a plethora of reaction–diffusion models were developed and proposed over the years to describe different morphogen gradients [30–33]. Some notable examples are Bcd in the syncytial Drosophila embryo [34], Dpp in developing wing imaginal disc in Drosophila [19], Fgf8 in the gastrulating Danio rerio embryo [35], among other examples. Despite the controversy of whether reaction–diffusion models represent an effective or accurate description of tissue pattern formation, these modelling frameworks became an essential construct to guide mathematical approaches in development [5,36] and regeneration [37].

In this study, we investigated the spatio-temporal distribution of a morphogen with a minimal reaction–diffusion model in a finite domain, as a proxy for a tissue. The solution of the model assuming an infinite domain has been already reported [10,11]. A number of reaction–diffusion models were previously considered to investigate morphogen gradients in finite domains, by means of numerical simulations (see, for instance, [7,12], among other examples). A reaction–diffusion model assuming finite domains was exactly solved assuming Neumann boundary conditions to investigate the scaling of morphogens in tissues by Umulis [13] and robustness of pattern formation in development [14], among other examples. A similar model was considered to investigate cell migration and proliferation of a population of precursor cells on a uniformly growing tissue by Simpson [38], based on the model of cell colonization in uniformly growing domains [39]. In his

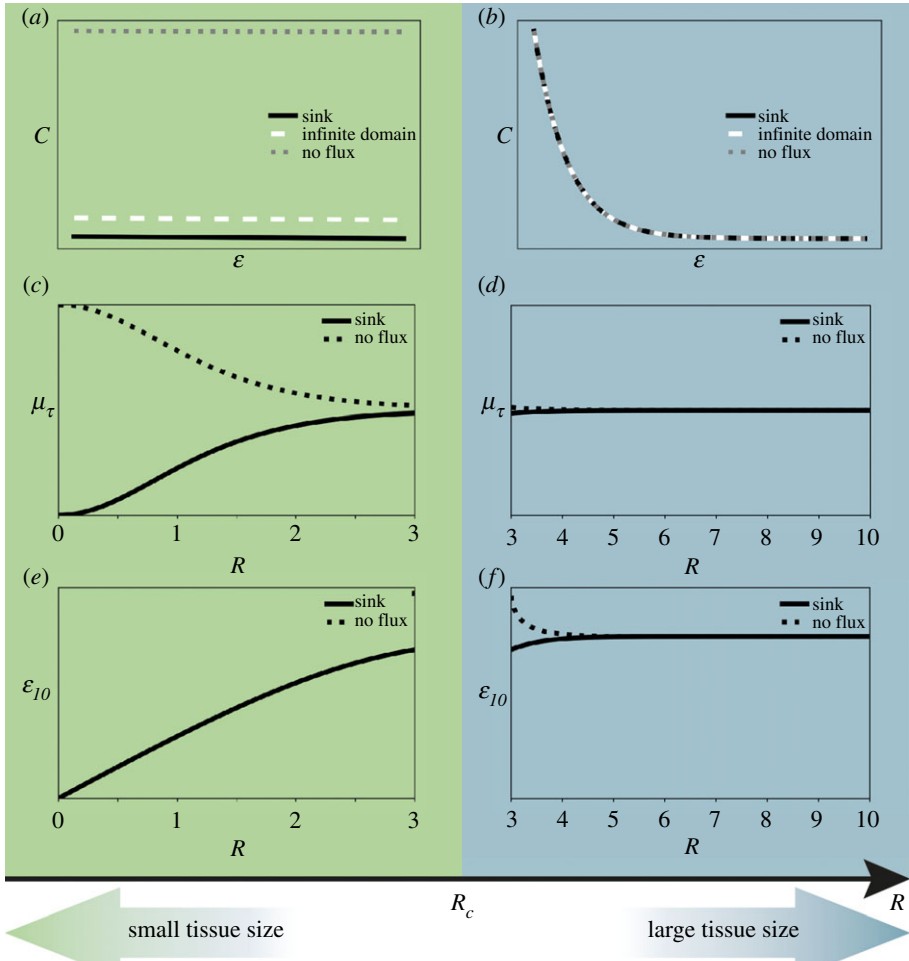

**Figure 7.** Transition between small and large tissues: two reaction–diffusion regimes separated by a crossover tissue size. Sketch summarizing the main differences between small and large tissue sizes separated by a crossover tissue size ($R_c$). Spatial profiles of morphogen concentration $C$ (a,b) and dependency of the geometrical factor $\varepsilon_{10}$ (c,d) and the mean time to reach the steady state $\mu_\tau$ (e,f) with the nondimensionalized tissue size $R$, for small (a,c,e) and large tissues (b,d,f).

model, Simpson [38] explored a more general case of a growing domain, which can recapitulate the case of a fixed domain by setting the growth speed to zero. Nevertheless, because the model focused on cells instead of morphogens, it assumed a positive reaction term to account for cell proliferation and a non-zero initial condition, in contrast with our negative reaction term and our zero initial condition. Hence, imposing a zero initial condition in this previously reported model yields the null solution.

The analytical solution here reported could be instrumental in computational packages devoted to multi-scale modelling, which involve a signalling scale coupled with a cellular scale. Although their cellular layer could entail a cellular Potts model [40,41] in CompuCell3D [42] and MORPHEUS [43], or a vertex model [44,45] in CHASTE [46], their signalling scale is typically modelled by a reaction–diffusion scheme. Since in these packages a finite domain is the only possible choice, they use numerical implementation. While our numerical results, based on a finite-difference algorithm, cannot be distinguished from the analytical solution (electronic supplementary material, figures S1 and S2), the last one is naturally more accurate and computationally more efficient (see electronic supplementary material), which could prove useful for multi-scale modelling implementations in one, two (see two examples of the two-dimensional finite-domain model in the electronic supplementary material) or three dimensions. Likewise, this new solution could help to improve the calculation of recovery curves in FRAP experiments, as for tissues below the crossover size $R_c$, the model assuming finite domains is a better approximation.

Our results showed that the morphogen spatial distributions predicted by our model assuming finite domains depend on the only relevant model parameter: the normalized tissue size $R$. By determining the spatial position along with the tissue where the morphogen concentration is 10% of the source concentration ($\varepsilon_{10}$), we geometrically characterized the steady state spatial distribution. This

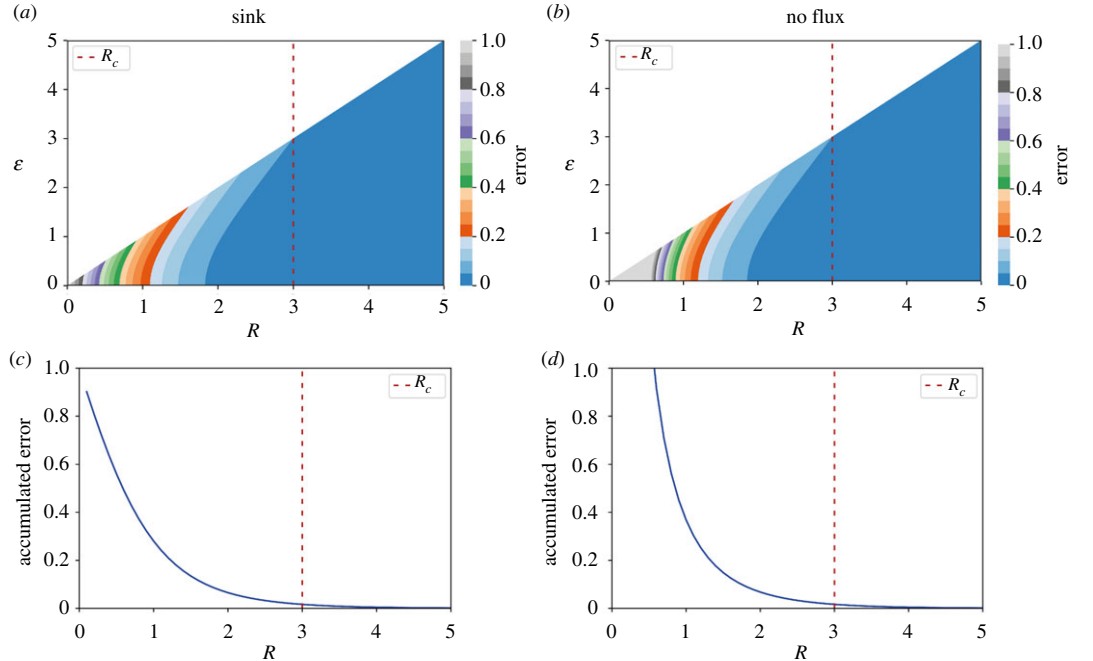

**Figure 8.** The reaction–diffusion model assuming a finite domain is a better approximation than the model assuming an infinite domain when the tissue size is smaller than the crossover tissue size. (a,b) Heat map showing the difference of morphogen concentration predicted from the reaction–diffusion finite-domain model and the infinite-domain model as a function of the nondimensionalized position within the tissue ($\varepsilon$) and the nondimensionalized tissue size ($R$). This difference could be considered as the error committed when using the model assuming the infinite-domain at a given position $\varepsilon$ from a tissue of size $R$. (c,d) The difference in (a) and (b), integrated over the tissue and nondimensionalized with $R$, as a function of $R$, representing the global error of using the infinite-domain model when the tissue size is $R$. The finite-domain model was simulated by assuming a sink boundary condition (a,c) or a no flux boundary condition (b,d) at $\varepsilon = R$. The vertical dashed lines indicate the crossover tissue size ($R_c$).

characterization led us to find two regimes within the parameters space, separated by a crossover tissue size $R_c$ (figure 3 and figure 7c,d). For tissues longer than $R_c$, the distributions are exponential-like and cannot be distinguished from those predicted from the model assuming an infinite domain (figures 2 and 7b). In this regime of the parameter space, the mean and standard deviation of the time to reach the steady state (evaluated at the tissue origin) do not change much with the tissue size and converged towards the corresponding values from the model assuming an infinite domain (figures 5 and 7f). When comparing the morphogen concentrations predicted by both models, we found that the difference between them is mostly negligible (figure 8a,b). Hence, the model assuming an infinite domain can be considered a good approximation of the model assuming finite domains for tissue sizes larger than $R_c$.

By contrast, for tissues smaller than $R_c$, the distributions are clearly separated from those predicted with the model assuming an infinite domain (figures 2 and 7a). Furthermore, the time to reach the steady state strongly depends on the tissue size in this regime and the particular boundary condition at $\varepsilon = R$ (figures 5 and 7e). In particular, the error of using the model assuming an infinite domain increases when $\varepsilon$ tends to $R$ and the smaller the tissue the higher the error accumulated over the entire tissue (figure 8a,b) (see electronic supplementary material for details). Thus, our results indicate that to investigate tissues smaller than approximately three times the characteristic length $\lambda$, the model assuming finite domains should be used.

In this study, we have analysed two different boundary conditions at the tip of the tissue simulated with the finite-domain model. For tissues smaller than $R_c$, the choice of this boundary condition is determinant for the amplitude of the morphogen spatial distribution and the time to achieve the steady state. When assuming a sink boundary condition, we obtained concentration values below those predicted by the infinite-domain model (figures 1a,c and 2a). Thus, this boundary condition leads to a reduction in the time to reach the steady state, when compared to the infinite-domain model (figure 5a). By contrast, a no flux boundary condition leads to concentrations higher than those calculated when assuming an infinite domain (figure 1b,d and 2b), which comparatively increases the time to approach the steady state (figure 5b).

The crossover tissue size provides a straightforward criterion to decide when to use any of the two models presented here. As an example, the characteristic length of Wg was estimated in 6 µm in the *Drosophila* wing disc, where the tissue size was about 50 µm [19]. The resulting $R \sim 8 > R_c$ indicates that the model assuming an infinite domain is a reasonable approximation in this scenario. A similar conclusion can be drawn when studying Dpp in the *Drosophila* haltere. For this morphogen, the characteristic length and the tissue size can be estimated in approximately 10 and approximately 100 µm, respectively [7], which leads to $R \sim 10 > R_c$. By contrast, the last morphogen, Dpp, but in the *Drosophila* wing disc has a characteristic length of 20 µm [19] which implies $R \sim 2.5 < R_c$. As a consequence, the model assuming finite tissues is the most correct approximation to describe morphogen propagation in this scenario. Something similar occurs with Fgf8 in the *Danio rerio* embryo, whose characteristic length was estimated as 200 µm while the tissue size is about 200 µm [35], from which a $R \sim 1 < R_c$ can be calculated. By only looking at the previous examples, it is clear that there is no correlation between the model selection and the morphogen under study, since the same morphogen, Dpp dynamics is better explained with the model assuming finite domains in the *Drosophila* wing disc while in the *Drosophila* the model assuming an infinite domain is actually sufficient. The same lack of correlation can be observed between the model selection and the tissue of interest. Indeed, in the same tissue, *Drosophila* imaginal disc, Wg could be described with the model assuming an infinite domain while Dpp requires the most precise model of finite domains.

In conclusion, we found two reaction–diffusion regimes for large and small tissues, separated by a crossover tissue size. While above this crossover the infinite-domain model constitutes a good approximation, it breaks below this crossover, whereas the finite-domain model faithfully describes the entire parameter space. Further studies will be needed to unveil the spatio-temporal distribution of morphogens in tissues whose size is not fixed. Our finding of the delineated crossover tissue size could be instrumental to select the proper reaction–diffusion model in future studies aimed to address tissue morphogenesis and other relevant problems regarding pattern formation in biology and medicine.

## 3.1. Computational methods

In this article, the reaction–diffusion model assuming a finite domain and its comparison with the model assuming an infinite domain was studied. The analytical derivation of the reaction–diffusion model assuming a finite domain for different boundary conditions in one and two dimensions is presented in §1 in electronic supplementary material. Comparison between analytical and numerical solutions in one and two dimensions is described in the §2 in the electronic supplementary material. Steady state calculations, the geometrical characterization of the spatial distribution profiles given by $\varepsilon_{10}$ and the estimation of the crossover tissue size $R_c$ are shown in §3, 4 and 5 in the electronic supplementary material, respectively. Mean time to reach the steady state together with its standard deviation are in §6 of electronic supplementary material. Calculation of FRAP recovery curves from the one-dimensional finite-domain model is in §7 of the electronic supplementary material. Details on the error of assuming an infinite domain instead of a finite domain in the steady state solutions are in §8 in the electronic supplementary material. Finally, the efficiency of the analytical solution versus the numerical simulations is analysed in §9 of the electronic supplementary material.

All model calculations were encoded in Python 3.7.3 and performed using NumPy [47] and SciPy [48] while visualization was executed with matplotlib [49] and seaborn [50]. The source codes for all the calculations and figures were implemented in supplementary notebooks using Jupyter Notebook (http://jupyter.org/) and can be found at: http://doi.org/10.5281/zenodo.4421327 [51].

Data accessibility. As we wrote in the Computational Methods section, the source codes for all the calculations and figures are implemented in supplementary notebooks using Jupyter Notebook (http://jupyter.org/) and can be found at: http://doi.org/10.5281/zenodo.4421327 [51]. The supplementary information is provided as the electronic supplementary material [52].
Authors' contributions. A.S.C.: conceptualization, formal analysis, investigation, software, visualization, writing—original draft and writing—review and editing; A.B.: investigation, software, visualization and writing—review and editing; O.C.: conceptualization, formal analysis, funding acquisition, investigation, project administration, resources, software, supervision, writing—original draft and writing—review and editing. All authors gave final approval for publication and agreed to be held accountable for the work performed therein.
Competing interests. We declare we have no competing interests.
Funding. This study was funded by Fondo para la Investigación Científica y Tecnológica (grants PICT-2014-3469 and PICT-2017-2307).

**Acknowledgements.** We thank Diane Peurichard and Valeria Caliaro from the INRIA Paris - team MAMBA and the Laboratoire Jacques Louis Lions (LJLL)- Sorbonne Université, Juan José Gervasio from the University of La Plata, Fabian Rost from the Center for Molecular and Cellular Bioengineering (CMCB), Technische Universität Dresden and the SysBio members of the Chara lab for their invaluable comments on this study.

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
