## [Peer Review File · Royal Society Open Science]

Review History

RSOS-211112.R0 (Original submission)

Review form: Reviewer 1

Is the manuscript scientifically sound in its present form?

Yes

Are the interpretations and conclusions justified by the results?

No

Is the language acceptable?

Yes

Do you have any ethical concerns with this paper?

No

Have you any concerns about statistical analyses in this paper?

No

Recommendation?

Major revision is needed (please make suggestions in comments)

Comments to the Author(s)

Size matters: an analytical study on the role of tissue size in spatiotemporal distribution of morphogens unveils a transition between different Reaction-Diffusion regimes by Alberto S. Ceccarelli and Augusto Borges and Osvaldo Chara

In this paper Ceccarelli et al study the spatio-temporal evolution of a morphogen diffusing and decaying in a 1D domain. Particularly, they compare the predictions in a finite domain with the predictions obtained assuming that the domain is infinite. This is an interesting question worthy of investigation and the results are technically sound and explained clearly. This analysis could have potentially important implications for the interpretation of experimental measurements with FRAP and it could also help to select the appropriate approximation in formulation of biological models involving the spread of morphogens. However, for both purposes there are important (yet solvable) issues that cast doubt into the relevance of this work. These points and other minor observations are elaborated below.

Firstly, the authors limit their analysis, at least in the main text, to the study of the RD equation with zero initial concentration, a constant flow of morphogen at the origin and a vanishing concentration at the opposite end. These type of boundary conditions are not necessarily the most relevant in a realistic biological setting. Arguably, a no-flux boundary condition at the opposite end of the source could describe more realistically the confinement of a morphogen in a tissue or in a cell. Importantly, the boundary conditions in this particular problem have a major effect in the results. For example, statements like “morphogen concentrations predicted by the model assuming an infinite domain are higher than those predicted by the model assuming a finite domain” would not hold true with a no-flux boundary condition at R. Furthermore, many of the differences between the finite and infinite domain explored at length in the text and figures, and even some of formulas derived analytically, are dependent on the boundary conditions. Since the authors mention that that they have in fact derived the analytical solution with alternative boundary conditions, I would suggest that they include at least a discussion of how they compare with the solution already discussed. The crossover length and time to reach steady states with alternative BCs should also be included. In addition, it would be important that the authors discuss if this was not included in the main text due to the overlap with ref.13 (Umulis 2009) or if their formulation of the problem and derivation of a solution are different in any important way.

Secondly, the relevance of the present study for the interpretation of FRAP experiments is not justified by the contents of the main text. Section 2.7 is almost entirely devoted to demonstrate once again that the two alternative assumptions, namely an infinite vs finite domain, produce different steady state predictions in domain lengths below a certain threshold, and that this threshold is the crossover length obtained in previous sections. This result is just a rehashing of the previous results but derived in a more convoluted way, which in fact obscures a point that has already been firmly established in the preceding sections: The steady state concentrations predicted using the two alternative assumptions differ significantly for lengths $L < 3 \cdot \lambda$. This derivation is redundant given the contents of the previous sections. Instead, I would suggest that the authors attempt to demonstrate that the theoretical formulation under study is relevant to interpret FRAP essays. For example, it would be helpful to show that the present mathematical formulation is a reasonable description of a FRAP essay. In a typical FRAP essay, a circular or square region containing a fluorescently labeled molecule is bleached and the recovery of the fluorescence levels caused by the diffusion of molecule back into the bleached region allow to

estimate the diffusion rates and decay parameters. Again, the boundary conditions discussed in the main text, and in this case also the geometrical setup (with a single source of morphogen in one of the boundaries) are not necessarily a good description of a FRAP assay. Would it be possible to derive the analytical solution for a case that more closely represents it? In addition, finding examples in the literature in which the estimation of morphogen parameters could be improved with the finite domain assumption and even providing the improved estimations would go a long way to attract attention to this work (much like it is done in the Discussion section with the analysis of the validity of the infinite-domain assumption for specific processes involving FGF8 and Dpp).

These two are the main issues that I find would improve this work, other less important points are explained next.

Section 2.6 explores the differences in time to reach steady state in the infinite vs finite-domain scenarios. This is done introducing the mean time and the standard deviation of the time to reach the steady state. These variables are typically associated with stochastic processes, but since all the equations studied throughout the manuscript are deterministic, it is not entirely clear what they represent. The reader is referred to the Supplementary Material for the derivation of their analytical expressions. This is fine, but I would suggest to include a brief explanation of how they enter in the deterministic description.

In several instances it is described as remarkable that the the analytical predictions and numerical simulations match. This, rather than a remarkable result, is a reassuring feature that confirms that the analytical derivation are correct.

Related to this, in the Discussion it is said this work is valuable for numerical packages because it is more accurate and efficient. This is not very convincing, since the numerical solution of this type of simple equations can be made as accurate as to be virtually indistinguishable from the analytical solutions, and also because typically these packages are used to simulate complex problems in “D or 3D geometries, for which the simple 1D analytical solution is not useful.

Review form: Reviewer 2

Is the manuscript scientifically sound in its present form?

No

Are the interpretations and conclusions justified by the results?

No

Is the language acceptable?

Yes

Do you have any ethical concerns with this paper?

No

Have you any concerns about statistical analyses in this paper?

No

Recommendation?

Major revision is needed (please make suggestions in comments)

Comments to the Author(s)

How tissue size affects the diffusion dynamics of morphogens is an interesting problem. In this sense I find the analyses performed in this study a useful addition to the literature on mathematical modeling of development. The results on the "crossover tissue size" is particularly interesting. A major problem of the manuscript is that the authors seem to have confused the "reaction-diffusion" model with the "French flag" model. As far as I can tell, this study has dealt with only the properties of morphogen diffusion; there is no "reaction" component. The solution for the real "reaction-diffusion" dynamics would be very different from the solution for a simple diffusion-only mechanism. I assume there are two ways to fix this: either reanalyze everything with a reaction component (which could be much more challenging than the current analysis) or rewrite the manuscript to put the work in the context of the "French flag" model rather than the "reaction-diffusion" model.

Decision letter (RSOS-211112.R0)

Dear Dr Chara

The Editors assigned to your paper RSOS-211112 "Size matters: an analytical study on the role of tissue size in spatiotemporal distribution of morphogens unveils a transition between different react" have now received comments from reviewers and would like you to revise the paper in accordance with the reviewer comments and any comments from the Editors. Please note this decision does not guarantee eventual acceptance.

Please submit your revised manuscript and required files (see below) no later than 21 days from today's (ie 13-Sep-2021) date. Note: the ScholarOne system will 'lock' if submission of the revision is attempted 21 or more days after the deadline. If you do not think you will be able to meet this deadline please contact the editorial office immediately.

on behalf of Dr Jose Carrillo (Associate Editor) and Mark Chaplain (Subject Editor)
 openscience@royalsociety.org

Reviewer comments to Author:

Reviewer: 1

Comments to the Author(s)

Size matters: an analytical study on the role of tissue size in spatiotemporal distribution of morphogens unveils a transition between different Reaction-Diffusion regimes by Alberto S. Ceccarelli and Augusto Borges and Osvaldo Chara

In this paper Ceccarelli et al study the spatio-temporal evolution of a morphogen diffusing and decaying in a 1D domain. Particularly, they compare the predictions in a finite domain with the predictions obtained assuming that the domain is infinite. This is an interesting question worthy of investigation and the results are technically sound and explained clearly. This analysis could have potentially important implications for the interpretation of experimental measurements with FRAP and it could also help to select the appropriate approximation in formulation of biological models involving the spread of morphogens. However, for both purposes there are important (yet solvable) issues that cast doubt into the relevance of this work. These points and other minor observations are elaborated below.

Firstly, the authors limit their analysis, at least in the main text, to the study of the RD equation with zero initial concentration, a constant flow of morphogen at the origin and a vanishing concentration at the opposite end. These type of boundary conditions are not necessarily the most relevant in a realistic biological setting. Arguably, a no-flux boundary condition at the opposite end of the source could describe more realistically the confinement of a morphogen in a tissue or in a cell. Importantly, the boundary conditions in this particular problem have a major effect in the results. For example, statements like “morphogen concentrations predicted by the model assuming an infinite domain are higher than those predicted by the model assuming a finite domain” would not hold true with a no-flux boundary condition at R. Furthermore, many of the differences between the finite and infinite domain explored at length in the text and figures, and even some of formulas derived analytically, are dependent on the boundary conditions. Since the authors mention that that they have in fact derived the analytical solution with alternative boundary conditions, I would suggest that they include at least a discussion of how they compare with the solution already discussed. The crossover length and time to reach steady states with alternative BCs should also be included. In addition, it would be important that the authors discuss if this was not included in the main text due to the overlap with ref.13 (Umulis 2009) or if their formulation of the problem and derivation of a solution are different in any important way.

Secondly, the relevance of the present study for the interpretation of FRAP experiments is not justified by the contents of the main text. Section 2.7 is almost entirely devoted to demonstrate once again that the two alternative assumptions, namely an infinite vs finite domain, produce different steady state predictions in domain lengths below a certain threshold, and that this threshold is the crossover length obtained in previous sections. This result is just a rehashing of the previous results but derived in a more convoluted way, which in fact obscures a point that has already been firmly established in the preceding sections: The steady state concentrations predicted using the two alternative assumptions differ significantly for lengths $L < 3 \cdot \lambda$. This derivation is redundant given the contents of the previous sections. Instead, I would suggest

that the authors attempt to demonstrate that the theoretical formulation under study is relevant to interpret FRAP essays. For example, it would be helpful to show that the present mathematical formulation is a reasonable description of a FRAP essay. In a typical FRAP essay, a circular or square region containing a fluorescently labeled molecule is bleached and the recovery of the fluorescence levels caused by the diffusion of molecule back into the bleached region allow to estimate the diffusion rates and decay parameters. Again, the boundary conditions discussed in the main text, and in this case also the geometrical setup (with a single source of morphogen in one of the boundaries) are not necessarily a good description of a FRAP essay. Would it be possible to derive the analytical solution for a case that more closely represents it? In addition, finding examples in the literature in which the estimation of morphogen parameters could be improved with the finite domain assumption and even providing the improved estimations would go a long way to attract attention to this work (much like it is done in the Discussion section with the analysis of the validity of the infinite-domain assumption for specific processes involving FGF8 and Dpp).

These two are the main issues that I find would improve this work, other less important points are explained next.

Section 2.6 explores the differences in time to reach steady state in the infinite vs finite-domain scenarios. This is done introducing the mean time and the standard deviation of the time to reach the steady state. These variables are typically associated with stochastic processes, but since all the equations studied throughout the manuscript are deterministic, it is not entirely clear what they represent. The reader is referred to the Supplementary Material for the derivation of their analytical expressions. This is fine, but I would suggest to include a brief explanation of how they enter in the deterministic description.

In several instances it is described as remarkable that the the analytical predictions and numerical simulations match. This, rather than a remarkable result, is a reassuring feature that confirms that the analytical derivation are correct.

Related to this, in the Discussion it is said this work is valuable for numerical packages because it is more accurate and efficient. This is not very convincing, since the numerical solution of this type of simple equations can be made as accurate as to be virtually indistinguishable from the analytical solutions, and also because typically these packages are used to simulate complex problems in "D or 3D geometries, for which the simple 1D analytical solution is not useful.

Reviewer: 2

Comments to the Author(s)

How tissue size affects the diffusion dynamics of morphogens is an interesting problem. In this sense I find the analyses performed in this study a useful addition to the literature on mathematical modeling of development. The results on the "crossover tissue size" is particularly interesting. A major problem of the manuscript is that the authors seem to have confused the "reaction-diffusion" model with the "French flag" model. As far as I can tell, this study has dealt with only the properties of morphogen diffusion; there is no "reaction" component. The solution for the real "reaction-diffusion" dynamics would be very different from the solution for a simple diffusion-only mechanism. I assume there are two ways to fix this: either reanalyze everything with a reaction component (which could be much more challenging than the current analysis) or rewrite the manuscript to put the work in the context of the "French flag" model rather than the "reaction-diffusion" model.

===PREPARING YOUR MANUSCRIPT===

===PREPARING YOUR REVISION IN SCHOLARONE===

- An individual file of each figure (EPS or print-quality PDF preferred [either format should be produced directly from original creation package], or original software format).
- An editable file of each table (.doc, .docx, .xls, .xlsx, or .csv).
- An editable file of all figure and table captions.

- Any electronic supplementary material (ESM).
- If you are requesting a discretionary waiver for the article processing charge, the waiver form must be included at this step.
- If you are providing image files for potential cover images, please upload these at this step, and inform the editorial office you have done so. You must hold the copyright to any image provided.
- A copy of your point-by-point response to referees and Editors. This will expedite the preparation of your proof.

- Ensure that your data access statement meets the requirements at <https://royalsociety.org/journals/authors/author-guidelines/#data>. You should ensure that you cite the dataset in your reference list. If you have deposited data etc in the Dryad repository, please include both the 'For publication' link and 'For review' link at this stage.
- If you are requesting an article processing charge waiver, you must select the relevant waiver option (if requesting a discretionary waiver, the form should have been uploaded at Step 3 'File upload' above).
- If you have uploaded ESM files, please ensure you follow the guidance at <https://royalsociety.org/journals/authors/author-guidelines/#supplementary-material> to include a suitable title and informative caption. An example of appropriate titling and captioning may be found at https://figshare.com/articles/Table_S2_from_Is_there_a_trade-off_between_peak_performance_and_performance_breadth_across_temperatures_for_aerobic_scope_in_teleost_fishes_/3843624.

Author's Response to Decision Letter for (RSOS-211112.R0)

See Appendix A.

RSOS-211112.R1 (Revision)

Review form: Reviewer 1

Is the manuscript scientifically sound in its present form?

Yes

Are the interpretations and conclusions justified by the results?

Yes

Is the language acceptable?

Yes

Do you have any ethical concerns with this paper?

No

Have you any concerns about statistical analyses in this paper?

No

Recommendation?

Accept with minor revision (please list in comments)

Comments to the Author(s)

The authors have addressed all my major concerns.

I would suggest to add a short paragraph in the discussion describing in plain words the major differences in behavior between the alternative assumptions (Dirichlet vs Neumann), like the time to reach steady state, the shape of the steady state, how they depart from the infinite-length solution and so on.

There is a statement in in pag. 64 that I find surprising: "With this boundary condition (no-flux), the total amount of morphogen accumulated in the tissue at the 218 steady state (NSS) is conserved and consequently, independent of R." Is this true, given that there is a decay term?

- The FRAP section is in much better shape. The authors tackled the problem of a bleached gradient, which is probably a much harder problem than what a typical FRAP essay entails. In a typical FRAP essay, a uniform distribution of a molecule is bleached. Since this is just a particular case of the more general scenario that they have solved, perhaps I'd be fitting if they briefly discuss it.

Aside from that, and to finish on a positive note, I'd like to congratulate the authors for this interesting piece of work.

Review form: Reviewer 2

Is the manuscript scientifically sound in its present form?

Yes

Are the interpretations and conclusions justified by the results?

Yes

Is the language acceptable?

Yes

Do you have any ethical concerns with this paper?

No

Have you any concerns about statistical analyses in this paper?

No

Recommendation?

Accept with minor revision (please list in comments)

Comments to the Author(s)

I appreciate the substantial effort that went into the revision of this manuscript, which made it an even stronger paper. I maintain my initial enthusiasm about the study, but I want to encourage the authors to think again about choosing the term "reaction-diffusion" over the more accurate alternatives such as SDD, Diffusion-decay, or Diffusion-degradation. The second term in the equation ($-kC$) describes the degradation of the same (diffusible) morphogen, not a "reaction" between two morphogens. If one mentions the term "reaction-diffusion model" to a developmental biologist, they will immediately think about two (or more) morphogens interacting with each other like in the "activator-inhibitor" or "activator-substrate depletion" systems. I assume the ultimate audience of this study are developmental biologists who are interested in mechanisms of pattern formation. Using a term that potentially confuses them will likely reduce the impact of the work. In both the "French flag" model and the real "Reaction-Diffusion" model, the properties of morphogen diffusion, be it over finite or infinite domain, are critically important. Therefore, choosing a more accurate but perhaps not as buzzing a term is not going to devalue the work. With that said, I am not demanding any changes. I'll leave this suggestion to the authors to consider.

Decision letter (RSOS-21112.R1)

Dear Dr Chara

On behalf of the Editors, we are pleased to inform you that your Manuscript RSOS-21112.R1 "Size matters: Tissue size as a marker for a transition between Reaction-Diffusion regimes in spatiotemporal distribution of morphogens" has been accepted for publication in Royal Society Open Science subject to minor revision in accordance with the referees' reports. Please find the referees' comments along with any feedback from the Editors below my signature.

Please submit your revised manuscript and required files (see below) no later than 7 days from today's (ie 13-Dec-2021) date. Note: the ScholarOne system will 'lock' if submission of the revision is attempted 7 or more days after the deadline. If you do not think you will be able to meet this deadline please contact the editorial office immediately.

on behalf of Dr Jose Carrillo (Associate Editor) and Mark Chaplain (Subject Editor)
 openscience@royalsociety.org

Reviewer comments to Author:

Reviewer: 2

Comments to the Author(s)

I appreciate the substantial effort that went into the revision of this manuscript, which made it an even stronger paper. I maintain my initial enthusiasm about the study, but I want to encourage the authors to think again about choosing the term "reaction-diffusion" over the more accurate alternatives such as SDD, Diffusion-decay, or Diffusion-degradation. The second term in the equation ($-kC$) describes the degradation of the same (diffusible) morphogen, not a "reaction" between two morphogens. If one mentions the term "reaction-diffusion model" to a developmental biologist, they will immediately think about two (or more) morphogens interacting with each other like in the "activator-inhibitor" or "activator-substrate depletion" systems. I assume the ultimate audience of this study are developmental biologists who are interested in mechanisms of pattern formation. Using a term that potentially confuses them will likely reduce the impact of the work. In both the "French flag" model and the real "Reaction-Diffusion" model, the properties of morphogen diffusion, be it over finite or infinite domain, are critically important. Therefore, choosing a more accurate but perhaps not as buzzing a term is not going to devalue the work. With that said, I am not demanding any changes. I'll leave this suggestion to the authors to consider.

Reviewer: 1

Comments to the Author(s)

The authors have addressed all my major concerns.

I would suggest to add a short paragraph in the discussion describing in plain words the major differences in behavior between the alternative assumptions (Dirichlet vs Neumann), like the time to reach steady state, the shape of the steady state, how they depart from the infinite-length solution and so on.

There is a statement in in pag. 64 that I find surprising: "With this boundary condition (no-flux), the total amount of morphogen accumulated in the tissue at the 218 steady state (NSS) is conserved and consequently, independent of R." Is this true, given that there is a decay term?

- The FRAP section is in much better shape. The authors tackled the problem of a bleached gradient, which is probably a much harder problem than what a typical FRAP essay entails. In a typical FRAP essay, a uniform distribution of a molecule is bleached. Since this is just a particular case of the more general scenario that they have solved, perhaps I'd be fitting if they briefly discuss it.

Aside from that, and to finish on a positive note, I'd like to congratulate the authors for this interesting piece of work.

===PREPARING YOUR MANUSCRIPT===

one version should clearly identify all the changes that have been made (for instance, in coloured highlight, in bold text, or tracked changes);

===PREPARING YOUR REVISION IN SCHOLARONE===

-- If you are requesting an article processing charge waiver, you must select the relevant waiver option (if requesting a discretionary waiver, the form should have been uploaded, see 'File upload' above).

-- If you have uploaded any electronic supplementary (ESM) files, please ensure you follow the guidance at <https://royalsociety.org/journals/authors/author-guidelines/#supplementary-material> to include a suitable title and informative caption. An example of appropriate titling and captioning may be found at https://figshare.com/articles/Table_S2_from_Is_there_a_trade-off_between_peak_performance_and_performance_breadth_across_temperatures_for_aerobic_scope_in_teleost_fishes_/3843624.

Author's Response to Decision Letter for (RSOS-211112.R1)

See Appendix B.

Decision letter (RSOS-211112.R2)

Dear Dr Chara,

I am pleased to inform you that your manuscript entitled "Size matters: Tissue size as a marker for a transition between Reaction-Diffusion regimes in spatiotemporal distribution of morphogens" is now accepted for publication in Royal Society Open Science.

Please remember to make any datasets or code libraries 'live' prior to publication, and update any links as needed when you receive a proof to check - for instance, from a private 'for review' URL to a publicly accessible 'for publication' URL. It is good practice to also add data sets, code and other digital materials to your reference list.

on behalf of Dr Jose Carrillo (Associate Editor) and Mark Chaplain (Subject Editor)
openscience@royalsociety.org

Appendix A

**TECHNISCHE
UNIVERSITÄT
DRESDEN**

Center for Information Services and High Performance Computing

Technische Universität Dresden, 01062 Dresden, Germany

Prof. Dr. Osvaldo Chara

Phone: 0351 463-38780
Fax: 0351 463-38245
Email: osvaldo.chara@tu-dresden.de
Web: www.zih.tu-dresden.de
Date: November 13th, 2021

Dresden, November 13th, 2021

Dear Editor,

we would like to thank you and the reviewers for the prompt and thoughtful review. The very constructive suggestions and comments of the reviewers allowed us to build, in our opinion, a more complete and thus stronger article. Essentially, the main elements of this rebuttal are as follows:

We included in our study the analysis of the no-flux boundary condition at the opposite end of the source, as suggested by the first reviewer. We performed the geometrical characterization of the morphogen gradients together with the calculation of the mean time to achieve the steady state predicted by our model with the no-flux boundary condition, as we did with the sink boundary condition presented in our original submission. The new results show differences between the sink and the no-flux boundary conditions. Noteworthy, the main result of the study holds: the existence of a crossover tissue length separating two reaction-diffusion regimes, regardless of the particular characteristics of the boundary conditions studied.

Thanks also to a suggestion of the first reviewer, we reformulated the section devoted to Fluorescence Recovery After Photobleaching (FRAP), where we recreated a simplified FRAP scenario. Our new results show that while a reaction-diffusion model assuming an infinite-domain could be used to study tissues larger than the crossover tissue length, the here proposed finite-domain model should be used for smaller tissues, in agreement with the main body of our results.

We also included a comparison between the model and numerical simulations in 2D in the supplementary information.

Finally, and thanks to the second reviewer, we improved the description of the model, making it, in our eyes, clearer.

Please find below a point-by-point response where we describe the additional results and changes in text. We look forward to your response.

Sincerely,

Osvaldo Chara

Postadresse (Briefe)

TU Dresden, 01062 Dresden
Postadresse (Pakete u.ä.)
TU Dresden
Helmholtzstraße 10
01069 Dresden

Besucheradresse

Sekretariat:
Musterstr. 1
Zimmer 1

Steuernummer

(Inland)
203/149/02549
Umsatzsteuer-Id-Nr.
(Ausland)
DE 188 369 991

Bankverbindung

Commerzbank AG,
Filiale Dresden
IBAN
DE52 8504 0000 0800 4004 00
BIC COBADEFF850

Zufahrt
Rampe Seiteneingang, gekennzeichnet.
Parkfläche im Innenhof

Internet

<http://tu-dresden.de>

Mitglied von:

**DRESDEN
Concept**
Exzellenz aus
Wissenschaft
und Kultur

Reviewer comments to Author:

Reviewer: 1

Comments to the Author(s)

Size matters: an analytical study on the role of tissue size in spatiotemporal distribution of morphogens unveils a transition between different Reaction-Diffusion regimes by Alberto S. Ceccarelli and Augusto Borges and Osvaldo Chara

In this paper Ceccarelli et al study the spatio-temporal evolution of a morphogen diffusing and decaying in a 1D domain. Particularly, they compare the predictions in a finite domain with the predictions obtained assuming that the domain is infinite. This is an interesting question worthy of investigation and the results are technically sound and explained clearly. This analysis could have potentially important implications for the interpretation of experimental measurements with FRAP and it could also help to select the appropriate approximation in formulation of biological models involving the spread of morphogens. However, for both purposes there are important (yet solvable) issues that cast doubt into the relevance of this work. These points and other minor observations are elaborated below.

Firstly, the authors limit their analysis, at least in the main text, to the study of the RD equation with zero initial concentration, a constant flow of morphogen at the origin and a vanishing concentration at the opposite end. These type of boundary conditions are not necessarily the most relevant in a realistic biological setting. Arguably, a no-flux boundary condition at the opposite end of the source could describe more realistically the confinement of a morphogen in a tissue or in a cell. Importantly, the boundary conditions in this particular problem have a major effect in the results. For example, statements like “morphogen concentrations predicted by the model assuming an infinite domain are higher than those predicted by the model assuming a finite domain” would not hold true with a no-flux boundary condition at R. Furthermore, many of the differences between the finite and infinite domain explored at length in the text and figures, and even some of formulas derived analytically, are dependent on the boundary conditions. Since the authors mention that that they have in fact derived the analytical solution with alternative boundary conditions, I would suggest that they include at least a discussion of how they compare with the solution already discussed. The crossover length and time to reach steady states with alternative BCs should also be included. In addition, it would be important that the authors discuss if this was not included in the main text due to the overlap with ref.13 (Umulis 2009) or if their formulation of the problem and derivation of a solution are different in any important way.

Answer: We thank the reviewer for her/his commentaries and the very constructive criticism. We fully agree with the suggestion of incorporating the no-flux boundary condition of the reaction-diffusion finite-domain model in our study. Thus, we decided to add the solution in the new version of the manuscript. We then extended the geometric characterization of the morphogen gradient together with the calculation of the main time to establish the steady state to this particular boundary condition. We were also able to define a crossover length for the model with the no-flux boundary condition which was similar to the one calculated from the sink boundary condition. As

Postadresse (Briefe)
TU Dresden, 01062 Dresden
Postadresse (Pakete u.ä.)
TU Dresden
Helmholtzstraße 10
01069 Dresden

Besucheradresse
Sekretariat:
Musterstr. 1
Zimmer 1
Steuernummer
(Inland)
203/149/02549
Umsatzsteuer-Id-Nr.
(Ausland)
DE 188 369 991

Bankverbindung
Commerzbank AG,
Filiale Dresden
IBAN
DE52 8504 0000 0800 4004 00
BIC COBADEFF850

Zufahrt
Rampe Seiteneingang, gekennzeichnet.
Parkfläche im Innenhof 2
Internet
<http://tu-dresden.de>

Mitglied von:
DRESDEN
Concept
Exzellenz aus
Wissenschaft
und Kultur

with the sink boundary condition, we observed that although the infinite-domain model cannot be distinguished from the finite-domain model for tissue lengths higher than the crossover tissue length, the morphogen gradients predicted by both models are clearly different for tissues smaller than the crossover length. Interestingly, and more prominently for these small tissues, the choice of the boundary condition at the opposite end of the source determines whether the steady state concentrations are higher or smaller than the concentrations predicted by the infinite-domain model. Because of that, while the sink boundary condition in the finite-domain model leads to values of the mean time to reach the steady state smaller than those predicted by the infinite-domain model, the opposite occurs with the no-flux boundary condition. In our eyes, the reviewer suggestion helped to build a more cohesive study comparing the finite-domain with the infinite-domain reaction-diffusion models.

As for the study of Umulis (2009), we acknowledge in our article the fact that our solution with the no-flux boundary condition coincides with the one previously obtained by him (reference 13). Nevertheless, as we mentioned in the discussion section, his study focused on the very interesting but completely different problem of morphogen scaling (which is reflected on the fact that he normalized space in units of the tissue length $\varepsilon = \frac{x}{L}$; in such units the effect of tissue size that we are interested in our study cannot be easily studied).

Secondly, the relevance of the present study for the interpretation of FRAP experiments is not justified by the contents of the main text. Section 2.7 is almost entirely devoted to demonstrate once again that the two alternative assumptions, namely an infinite vs finite domain, produce different steady state predictions in domain lengths below a certain threshold, and that this threshold is the crossover length obtained in previous sections. This result is just a rehashing of the previous results but derived in a more convoluted way, which in fact obscures a point that has already been firmly established in the preceding sections: The steady state concentrations predicted using the two alternative assumptions differ significantly for lengths $L < 3 \cdot \lambda$. This derivation is redundant given the contents of the previous sections. Instead, I would suggest that the authors attempt to demonstrate that the theoretical formulation under study is relevant to interpret FRAP essays. For example, it would be helpful to show that the present mathematical formulation is a reasonable description of a FRAP essay. In a typical FRAP essay, a circular or square region containing a fluorescently labeled molecule is bleached and the recovery of the fluorescence levels caused by the diffusion of molecule back into the bleached region allow to estimate the diffusion rates and decay parameters. Again, the boundary conditions discussed in the main text, and in this case also the geometrical setup (with a single source of morphogen in one of the boundaries) are not necessarily a good description of a FRAP essay. Would it be possible to derive the analytical solution for a case that more closely represents it? In addition, finding examples in the literature in which the estimation of morphogen parameters could be improved with the finite domain assumption and even providing the improved estimations would go a long way to attract attention to this work (much like it is done

Postadresse (Briefe)
TU Dresden, 01062 Dresden
Postadresse (Pakete u.ä.)
TU Dresden
Helmholtzstraße 10
01069 Dresden

Besucheradresse
Sekretariat:
Musterstr. 1
Zimmer 1

Steuernummer
(Inland)
203/149/02549
Umsatzsteuer-Id-Nr.
(Ausland)
DE 188 369 991

Bankverbindung
Commerzbank AG,
Filiale Dresden
IBAN
DE52 8504 0000 0800 4004 00
BIC COBADEFF850

Zufahrt
Rampe Seiteneingang, gekennzeichnet.
Parkfläche im Innenhof 3

Internet
<http://tu-dresden.de>

Mitglied von:

DRESDEN
Concept
Exzellenz aus
Wissenschaft
und Kultur

in the Discussion section with the analysis of the validity of the infinite-domain assumption for specific processes involving FGF8 and Dpp).

Answer: We thank the reviewer for her/his great suggestion. We completely reformulated the FRAP section of our manuscript. We simulated a typical FRAP experiment with the finite-domain model (with either the sink or the no-flux boundary condition at the opposite end of the source) and calculated the FRAP recovery curves for different domain sizes to then compare them with the recovery curve assuming an infinite domain, previously reported by Kicheva and colleagues (2007). We found that, consistently with our previous results, domains larger than the crossover length generate recovery curves indistinguishable from the one predicted by the infinite-domain model. In contrast, recovery curves smaller than the crossover length clearly differ from the infinite-domain model. Interestingly, the choice of the boundary condition at the opposite end of the source determines whether the recovery curves simulated with the finite-domain model are faster or slower than the one predicted by the infinite-domain model, in agreement with the results mentioned in our previous answer.

In the FRAP experiments, diffusion coefficient and half-life of morphogens is typically obtained through the fitting of fluorescence recovery data by an expression derived from the reaction-diffusion model. To that end, as a proof of principle, we evaluated whether the infinite domain model could render an accurate estimation of these parameters, when fitted to a dataset recovery curve, in turn generated with the finite domain model, used as a proxy for an experimental data recovery curve. We found that the infinite-domain model give rise to correct parameter estimations provided that tissue sizes are higher than crossover length. On the contrary, for tissues smaller than this crossover value, the parameter estimations depart from the correct values, indicating that the model assuming finite domains is the best alternative.

These two are the main issues that I find would improve this work, other less important points are explained next.

Section 2.6 explores the differences in time to reach steady state in the infinite vs finite-domain scenarios. This is done introducing the mean time and the standard deviation of the time to reach the steady state. These variables are typically associated with stochastic processes, but since all the equations studied throughout the manuscript are deterministic, it is not entirely clear what they represent. The reader is referred to the Supplementary Material for the derivation of their analytical expressions. This is fine, but I would suggest to include a brief explanation of how they enter in the deterministic description.

Answer: We thank the reviewer for the suggestion. We added a brief explanation on this point when introducing the methodology.

In several instances it is described as remarkable that the the analytical predictions and numerical simulations match. This, rather than a remarkable result, is a reassuring feature that confirms that the analytical derivation are correct.

Postadresse (Briefe)
TU Dresden, 01062 Dresden

Postadresse (Pakete u.ä.)
TU Dresden
Helmholtzstraße 10
01069 Dresden

Besucheradresse
Sekretariat:
Musterstr. 1
Zimmer 1

Steuernummer
(Inland)
203/149/02549

Umsatzsteuer-Id-Nr.
(Ausland)
DE 188 369 991

Bankverbindung
Commerzbank AG,
Filiale Dresden
IBAN
DE52 8504 0000 0800 4004 00
BIC COBADEFF850

Zufahrt
Rampe Seiteneingang, gekennzeichnet.
Parkfläche im Innenhof 4

Internet
<http://tu-dresden.de>

Mitglied von:

DRESDEN
Concept
Exzellenz aus
Wissenschaft
und Kultur

Answer: We agree with the reviewer and we corrected the text accordingly.

Related to this, in the Discussion it is said this work is valuable for numerical packages because it is more accurate and efficient. This is not very convincing, since the numerical solution of this type of simple equations can be made as accurate as to be virtually indistinguishable from the analytical solutions, and also because typically these packages are used to simulate complex problems in “D or 3D geometries, for which the simple 1D analytical solution is not useful.

Answer: We thank the reviewer for the suggestion. We added preliminary results in the supplementary information, where we compared the analytical with numerical solutions in 2D geometries.

Reviewer: 2

Comments to the Author(s)

How tissue size affects the diffusion dynamics of morphogens is an interesting problem. In this sense I find the analyses performed in this study a useful addition to the literature on mathematical modeling of development. The results on the "crossover tissue size" is particularly interesting. A major problem of the manuscript is that the authors seem to have confused the "reaction-diffusion" model with the "French flag" model. As far as I can tell, this study has dealt with only the properties of morphogen diffusion; there is no "reaction" component. The solution for the real "reaction-diffusion" dynamics would be very different from the solution for a simple diffusion-only mechanism. I assume there are two ways to fix this: either reanalyze everything with a reaction component (which could be much more challenging than the current analysis) or rewrite the manuscript to put the work in the context of the "French flag" model rather than the "reaction-diffusion" model.

Answer: We thank the reviewer for her/his commentaries and appreciate the interest on the problem and her/his appreciation of the section describing the ‘crossover tissue size’.

The name of the model we describe in our study varies depending on the author. For example, in Gregor *et al.*, 2007. Cell. 130(1): 141-152, the authors used the same model that we present, but with the name “*Synthesis, Diffusion, and Degradation model or SDD model*”. In Rasolonjanahary *et al.*, 2016. J Theor Biol. 404: 109-119, the authors called “*Diffusion-decay systems*” to the same model. Another example is Wartlick *et al.*, 2009. Cold Spring Harb Perspect Biol.1(3): a001255, where the authors used the same model, naming it as “*Diffusion Equation with linear degradation*”. In our study, we decided to refer to this model as “*Reaction-Diffusion model*” because it can be thought as the simplest case of the reaction-diffusion model as described by Alan Turing (see Turing *et al.*, 1952. Phil Trans R Soc Lond B. 237(641): 37-72). Thus, in our model, the first term of equation 1 of the revised version of the manuscript ($D \frac{d^2C}{dx^2}$) accounts for the diffusion process and the second term ($-kC$) corresponds to the ‘reaction’, which models a (linear) degradation or an uptake process. We are

Postadresse (Briefe)
TU Dresden, 01062 Dresden
Postadresse (Pakete u.ä.)
TU Dresden
Helmholtzstraße 10
01069 Dresden

Besucheradresse
Sekretariat:
Musterstr. 1
Zimmer 1

Steuernummer
(Inland)
203/149/02549
Umsatzsteuer-Id-Nr.
(Ausland)
DE 188 369 991

Bankverbindung
Commerzbank AG,
Filiale Dresden
IBAN
DE52 8504 0000 0800 4004 00
BIC COBADEFF850

Zufahrt
Rampe Seiteneingang, gekennzeichnet.
Parkfläche im Innenhof 5

Internet
<http://tu-dresden.de>

Mitglied von:

DRESDEN
concept
Exzellenz aus
Wissenschaft
und Kultur

aware that this term in general could be much more complicated. However, we precisely selected the simplest possible reaction term as a proof of principle. Thus, as the reviewer can see, the model does include a reaction component. However, following the reviewer's question, we decided to improve the description of the model differential equation when we first introduced it in the revised version of the manuscript.

Postadresse (Briefe)
TU Dresden, 01062 Dresden
Postadresse (Pakete u.ä.)
TU Dresden
Helmholtzstraße 10
01069 Dresden

Besucheradresse
Sekretariat:
Musterstr. 1
Zimmer 1

Steuernummer
(Inland)
203/149/02549
Umsatzsteuer-Id-Nr.
(Ausland)
DE 188 369 991

Bankverbindung
Commerzbank AG,
Filiale Dresden
IBAN
DE52 8504 0000 0800 4004 00
BIC COBADEFF850

Zufahrt
Rampe Seiten-
eingang, gekennzeichn.
Parkfläche im Innenhof 6
Internet
<http://tu-dresden.de>

Mitglied von:
DRESDEN
concept
Exzellenz aus
Wissenschaft
und Kultur

Technische Universität Dresden, 01062 Dresden, Germany

Prof. Dr. Osvaldo Chara

Phone: 0351 463-38780
Fax: 0351 463-38245
Email: osvaldo.chara@tu-dresden.de
Web: www.zih.tu-dresden.de
Date: December 20th, 2021

Dresden, December 20th, 2021

Dear Editor,

I am pleased that our manuscript entitled 'Size matters: Tissue size as a marker for a transition between Reaction-Diffusion regimes in spatiotemporal distribution of morphogens' (RSOS-211112.R1) has been accepted for publication in Royal Society Open Science subject to minor revision in accordance with the referees' reports.

In this final revised version, we have addressed all the suggestions from reviewer 1 and 2. Again, we would like to thank you and the reviewers for the very constructive suggestions and comments, which, as mentioned in my previous cover letter, allowed us to generate a more complete and thus stronger article. I believe that science is more often than not a collective enterprise and I find this article a modest example where all, authors, reviewers and editors collectively generated a scientific piece.

Please find below a point-by-point response where we describe the additional changes introduced in text. We look forward to your response.

Sincerely,

Osvaldo Chara

Reviewer comments to Author:

Reviewer: 2

Comments to the Author(s)

I appreciate the substantial effort that went into the revision of this manuscript, which made it an even stronger paper. I maintain my initial enthusiasm about the study, but I want to encourage the authors to think again about choosing the term "reaction-diffusion" over the more accurate alternatives such as SDD, Diffusion-decay, or Diffusion-degradation. The second term in the equation (-kC) describes the degradation of the same (diffusible) morphogen, not a "reaction" between two morphogens. If one mentions the term "reaction-diffusion model" to a developmental biologist, they will immediately think about two (or more) morphogens interacting with each other like in the "activator-inhibitor" or "activator-substrate depletion" systems. I assume the ultimate audience of

Postadresse (Briefe)

TU Dresden, 01062 Dresden
Postadresse (Pakete u.ä.)
TU Dresden
Helmholtzstraße 10
01069 Dresden

Besucheradresse

Sekretariat:
Musterstr. 1
Zimmer 1

Steuernummer

(Inland)
203/149/02549
Umsatzsteuer-Id-Nr.
(Ausland)
DE 188 369 991

Bankverbindung

Commerzbank AG,
Filiale Dresden
IBAN
DE52 8504 0000 0800 4004 00
BIC COBADEFF850

Zufahrt
Rampe Seiten-
eingang, gekennzeichn.
Parkfläche im Innenhof

Internet

<http://tu-dresden.de>

Mitglied von:

DRESDEN
concept
Exzellenz aus
Wissenschaft
und Kultur

this study are developmental biologists who are interested in mechanisms of pattern formation. Using a term that potentially confuses them will likely reduce the impact of the work. In both the "French flag" model and the real "Reaction-Diffusion" model, the properties of morphogen diffusion, be it over finite or infinite domain, are critically important. Therefore, choosing a more accurate but perhaps not as buzzing a term is not going to devalue the work. With that said, I am not demanding any changes. I'll leave this suggestion to the authors to consider.

Answer: We thank the reviewer for her/his suggestion. Following her/his advice, we included the three other alternative model names between lines 91 and 93, page 6 of the revised manuscript. We would like to take this opportunity to thank again the reviewer for the enthusiasm and helpful perspective.

Reviewer: 1

Comments to the Author(s)

The authors have addressed all my major concerns.

I would suggest to add a short paragraph in the discussion describing in plain words the major differences in behavior between the alternative assumptions (Dirichlet vs Neumann), like the time to reach steady state, the shape of the steady state, how they depart from the infinite-length solution and so on.

Answer: We thank the reviewer for the suggestion. Great idea! We added a short paragraph in the discussion addressing this point between lines 506 and 514, page 33 of the revised discussion section.

There is a statement in in pag. 64 that I find surprising: "With this boundary condition (no-flux), the total amount of morphogen accumulated in the tissue at the 218 steady state (NSS) is conserved and consequently, independent of R." Is this true, given that there is a decay term?

Answer: We thank the reviewer for the question. We agree that this result seems somehow surprising. Nevertheless, it is correct: when using the no-flux boundary condition, the integral of the steady state concentration over the normalized space is equal to 1 (Eq. 18). Thus, this integral, representing the total mass of accumulated morphogen, is conserved and, importantly, does not depend on the tissue/domain size. In contrast, when using the other boundary condition, this integral depends on the tissue size (although we did not show this last integral in the manuscript). We would also like to point out that, our sentence refers to the total morphogen mass accumulated at the steady state, where the decay term balances the source located at the origin.

Based on the reviewer's question, we added a brief comment between lines 213 and 218, page 14 of the revised Results section.

Postadresse (Briefe)
TU Dresden, 01062 Dresden
Postadresse (Pakete u.ä.)
TU Dresden
Helmholtzstraße 10
01069 Dresden

Besucheradresse
Sekretariat:
Musterstr. 1
Zimmer 1

Steuernummer
(Inland)
203/149/02549
Umsatzsteuer-Id-Nr.
(Ausland)
DE 188 369 991

Bankverbindung
Commerzbank AG,
Filiale Dresden
IBAN
DE52 8504 0000 0800 4004 00
BIC COBADEFF850

Zufahrt
Rampe Seiten-
eingang, gekennzeichn.
Parkfläche im Innenhof 2

Internet
<http://tu-dresden.de>

Mitglied von:

DRESDEN
Concept
Exzellenz aus
Wissenschaft
und Kultur

- The FRAP section is in much better shape. The authors tackled the problem of a bleached gradient, which is probably a much harder problem than what a typical FRAP essay entails. In a typical FRAP essay, a uniform distribution of a molecule is bleached. Since this is just a particular case of the more general scenario that they have solved, perhaps I'd be fitting if they briefly discuss it.

Answer: We thank the reviewer for the suggestion. We added a sentence between lines 370 and 373 in page 24 of the Results section commenting that although we investigated the bleached gradient to address the problem of a morphogen gradient, FRAP essays also involve uniform distributions.

Aside from that, and to finish on a positive note, I'd like to congratulate the authors for this interesting piece of work.

Answer: We want to thank again the reviewer for her/his detailed commentaries and positive criticism during this reviewing process!

Postadresse (Briefe)
TU Dresden, 01062 Dresden
Postadresse (Pakete u.ä.)
TU Dresden
Helmholtzstraße 10
01069 Dresden

Besucheradresse
Sekretariat:
Musterstr. 1
Zimmer 1

Steuernummer
(Inland)
203/149/02549
Umsatzsteuer-Id-Nr.
(Ausland)
DE 188 369 991

Bankverbindung
Commerzbank AG,
Filiale Dresden
IBAN
DE52 8504 0000 0800 4004 00
BIC COBADEFF850

Zufahrt
Rampe Seiteneingang, gekennzeichnet.
Parkfläche im Innenhof 3
Internet
<http://tu-dresden.de>

Mitglied von:
DRESDEN
concept
Exzellenz aus
Wissenschaft
und Kultur